# Population genomics confirms acquisition of drug-resistant *Aspergillus fumigatus* infection by humans from the environment

Johanna Rhodes [1]✉, Alireza Abdolrasouli[2,3], Katie Dunne[4], Thomas R. Sewell[1], Yuyi Zhang[1], Eloise Ballard[5], Amelie P. Brackin[1], Norman van Rhijn [6], Harry Chown [6], Alexandra Tsitsopoulou[7], Raquel B. Posso[8], Sanjay H. Chotirmall [9], Noel G. McElvaney[10], Philip G. Murphy [4,11], Alida Fe Talento [4], Julie Renwick[4], Paul S. Dyer[12], Adrien Szekely[13], Paul Bowyer[6], Michael J. Bromley [6], Elizabeth M. Johnson[13,14], P. Lewis White [8], Adilia Warris [5,14], Richard C. Barton [15], Silke Schelenz[16], Thomas R. Rogers[4], Darius Armstrong-James [2]✉ and Matthew C. Fisher [1]✉

**Infections caused by the fungal pathogen *Aspergillus fumigatus* are increasingly resistant to first-line azole antifungal drugs. However, despite its clinical importance, little is known about how susceptible patients acquire infection from drug-resistant genotypes in the environment. Here, we present a population genomic analysis of 218 *A. fumigatus* isolates from across the UK and Ireland (comprising 153 clinical isolates from 143 patients and 65 environmental isolates). First, phylogenomic analysis shows strong genetic structuring into two clades (A and B) with little interclade recombination and the majority of environmental azole resistance found within clade A. Second, we show occurrences where azole-resistant isolates of near-identical genotypes were obtained from both environmental and clinical sources, indicating with high confidence the infection of patients with resistant isolates transmitted from the environment. Third, genome-wide scans identified selective sweeps across multiple regions indicating a polygenic basis to the trait in some genetic backgrounds. These signatures of positive selection are seen for loci containing the canonical genes encoding fungicide resistance in the ergosterol biosynthetic pathway, while other regions under selection have no defined function. Lastly, pan-genome analysis identified genes linked to azole resistance and previously unknown resistance mechanisms. Understanding the environmental drivers and genetic basis of evolving fungal drug resistance needs urgent attention, especially in light of increasing numbers of patients with severe viral respiratory tract infections who are susceptible to opportunistic fungal superinfections.**

Fungal infections affect more than a billion people worldwide, with a mortality rate that matches that of malaria or tuberculosis[1]. A growing concern is *Aspergillus fumigatus*, a globally prevalent environmental mould that can cause multiple clinical diagnoses. Among these, invasive aspergillosis (IA) can occur in at-risk populations, such as patients with severe neutropenia, haematopoietic stem cell or solid organ transplants, patients receiving immunosuppressive drugs and, increasingly, patients with influenza and coronavirus disease 2019 (COVID-19) (as an associated infection)[2,3]. Patients with cystic fibrosis (CF) are also at risk of chronic infections, with 30% developing *Aspergillus*-related bronchitis and 19% developing allergic bronchopulmonary aspergillosis[4]. With over 2.25 million individuals suffering from infections caused by

*A. fumigatus* in the European Union alone[5], this is a global concern. Unfortunately, recent studies have also reported an emerging worldwide resistance to azole antifungal drugs in both clinical and environmental isolates[6–8], which have long proven effective against *A. fumigatus*[9].

Azole drug resistance has serious clinical implications, with retrospective studies of patients with drug-resistant IA showing a 25% increase in mortality at day 90 compared to patients with wild-type (WT) infections[10]. While in vivo resistance emergence during extended azole therapy is well documented[11,12], more recent studies postulate an ex vivo evolution of resistance in the environment as a result of exposure to agricultural chemicals—particularly sterol 14α-demethylation inhibitor fungicides developed in the

[1]Medical Research Council Centre for Global Disease Analysis, Imperial College London, London, UK. [2]Department of Infectious Diseases, Imperial College London, London, UK. [3]Department of Medical Microbiology, King's College University Hospital, London, UK. [4]Department of Clinical Microbiology, Trinity College Dublin, Dublin, Ireland. [5]Aberdeen Fungal Group, Institute of Medical Sciences, University of Aberdeen, Aberdeen, UK. [6]Manchester Fungal Infection Group, Faculty of Biology, Medicine and Health, The University of Manchester, Manchester Academic Health Science Centre, Core Technology Facility, Manchester, UK. [7]Microbiology Department, Royal Glamorgan Hospital, Cwm Taf NHS Trust, Ynysmaerdy, UK. [8]Public Health Wales Microbiology, Cardiff, UK. [9]Lee Kong Chian School of Medicine, Nanyang Technological University, Singapore, Singapore. [10]Respiratory Research Division, Department of Medicine, Royal College of Surgeons in Ireland, Education and Research Centre, Beaumont Hospital, Dublin, Ireland. [11]Department of Medical Microbiology, Tallaght University Hospital, Dublin, Ireland. [12]School of Life Sciences, University of Nottingham, Nottingham, UK. [13]Head Mycology Reference Laboratory, UK Health Security Agency, Bristol, UK. [14]Medical Research Council Centre for Medical Mycology, University of Exeter, Exeter, UK. [15]Mycology Reference Centre, Leeds Teaching Hospitals National Health Service Trust, Leeds, UK. [16]Infection Sciences, Kings College University Hospital, London, UK. ✉e-mail: johanna.rhodes@imperial.ac.uk; d.armstrong@imperial.ac.uk; matthew.fisher@imperial.ac.uk

1970s[13,14]. Broadly, environmentally occurring azole resistance in *A. fumigatus* is characterized by signature mechanisms involving expression-upregulating tandem repeats (TRs) in the promoter region of *cyp51A* accompanied by point mutations within this gene, which decrease the affinity of azoles for the target protein; the most commonly occurring alleles are known as TR$_{34}$/L98H and TR$_{46}$/Y121F/T289A and are associated with high-level itraconazole and voriconazole resistance, respectively both inside and outside the clinic[15–17]. The spatially widespread occurrence of these alleles alongside increasing reports of more complex *cyp51A* resistance-associated polymorphisms[18–20] underpin the hypothesis that the broad application of agricultural azole fungicides is driving natural selection, amplification and ultimately acquisition of azole-resistant airborne *A. fumigatus* conidia by susceptible patients[21]. Furthermore, the potential for global spread of these resistance mechanisms through floriculture products, especially plant bulbs, has been demonstrated[22], while the global dispersal of conidia on air currents is impossible to contain.

Modern genomic epidemiological methods further indicate a potential link between the increasing clinical incidence of azole-resistant IA and the increasingly broad range of azole-resistant genotypes that are being reported in the environment[23]. The rate at which environmental resistance develops will be determined by natural selection on beneficial mutations. This is in turn influenced by recombination, gene flow and dispersal, which leave their characteristic signatures in the genome. Evidence for these expectations comes from a recent global study from our laboratory demonstrating the non-random distribution of azole resistance in multilocus microsatellite genotypes[24].

In this study, we used whole-genome sequencing (WGS) of 218 *A. fumigatus* isolates ($n = 65$ environmental isolates and $n = 153$ clinical isolates) to interrogate the molecular epidemiology of this fungus and determine whether acquisition of drug-resistant isolates by at-risk patient groups occurs. We also leveraged the power of these data to perform genome-wide association studies (GWAS) and pan-genome analyses to identify the variation associated with itraconazole drug resistance, revealing potentially new mechanisms of resistance.

## Results

**WGS of *218 A. fumigatus* isolates.** In this study, we used reference-guided and de novo assembly methods to analyse the population genomics and pan-genome of 218 clinical and environmental *A. fumigatus* isolates, spanning the UK and Ireland (Supplementary Table 1). Of these 218 sequenced isolates, 153 (70%) were clinical in origin and the remaining 65 (30%) originated from environmental sources in the UK and Ireland. The genomic dataset can be accessed as a Microreact project[25] at https://microreact.org/project/6KR8996ywtVRV5wm233YhP (Extended Data Fig. 1). A chi-squared test showed a significant bias towards the *MAT1-2* idiomorph ($P < 4.82983 \times 10^{-05}$), primarily seen in the environmental ($P < 1.00183 \times 10^{-05}$) and to a lesser extent the clinical ($P < 0.04396$) populations (Supplementary Table 2). Only one isolate (ARAF005) displayed partial ploidy of chromosome 1.

Of the isolates tested, 106 (49%) showed resistance to at least 1 of the tested antifungal drugs. For specific azoles, 52% ($n = 104$) exceeded minimal inhibitory concentration (MIC) breakpoints to itraconazole ($\geq 2$ mg l$^{-1}$), 31% ($n = 64$) to voriconazole ($\geq 2$ mg l$^{-1}$) and 25% ($n = 44$) to posaconazole ($\geq 0.5$ mg l$^{-1}$). We found 26 isolates (12%) that exceeded MIC breakpoints to 2 or more azole drugs from both clinical ($n = 23$) and environmental ($n = 3$) sources. Seventeen isolates were not tested for antifungal susceptibility via European Committee on Antimicrobial Susceptibility Testing (EUCAST) or Clinical and Laboratory Standards Institute (CLSI); of these, 14 contained drug resistance polymorphisms (TR$_{34}$/L98H or TR$_{46}$/Y121F/T289A) and 3 contained no known drug resistance polymorphism.

Thirteen isolates reported raised MICs but displayed no known drug resistance polymorphisms (Supplementary Table 3).

A total of 329,261 sites were polymorphic (approximately 1.1%), a proportion that was in line with our previous study[21]. Pairwise identities showed that, on average, each isolate of *A. fumigatus* in this dataset differed from all others by 11,828 single-nucleotide polymorphisms (SNPs). Genotypes and their relative frequencies are shown in Supplementary Table 4.

**Phylogenomics and signatures of selection associated with resistance.** Phylogenetic analysis identified two broadly divergent clades (Fig. 1 and Supplementary Fig. 2) with 100% bootstrap support; 'clade A' and 'clade B' contained 123 and 95 isolates, respectively (Supplementary Table 5). An additional 41 publicly available WGS of non-UK origin were added to the phylogeny to confirm that clade assignment (Extended Data Fig. 3 and Supplementary Table 6) was not an artefact of these data but also seen globally. Most ($n = 99$) resistance-associated genotypes and azole-resistant phenotypes clustered within clade A (Fig. 1, Supplementary Fig. 2 and Supplementary Table 5). Conversely, most isolates with no resistance polymorphisms and azole-susceptible phenotypes ($n = 82$) clustered into clade B. MICs for three azole drugs, as well as the occurrence of polymorphisms associated with the *cyp51A* gene, were significantly higher in clade A than in clade B (chi-squared test $P = 3.27308 \times 10^{-14}$, d.f. = 1). For TR-associated polymorphisms, 100% of the TR$_{34}$ and 71% ($n = 5$) of the TR$_{46}$-associated alleles occurred within clade A.

Twenty-three per cent ($n = 75,317$ SNPs) of the total *A. fumigatus* diversity seen in the dataset occurred within clade A, despite comprising 56% of the isolates sampled. No SNPs were uniquely associated with the TR$_{34}$/L98H or TR$_{46}$ polymorphisms. This mirrors earlier STR*Af* microsatellite analysis, illustrating that isolates harbouring drug resistance polymorphisms displayed reduced genetic diversity and were genetically depauperate compared to randomly selected WT isolates[24]. Multivariate methods were used to identify and describe clusters of genetically related isolates; principal component analysis (PCA) identified three optimal clusters, with no geographical or temporal clustering, which broadly corresponded to the phylogeny (Extended Data Fig. 4). Cluster 1 corresponded to a subset of clade A containing a broad selection of *cyp51A* polymorphisms; cluster 2 corresponded to clade B; and cluster 3 overlapped with isolates within clade A containing TR$_{34}$/L98H only. Discriminant PCA and STRUCTURE confirmed the three clusters (Fig. 2b,c). Estimates of index of association and rBarD implied no significant linkage among the loci for all three clusters, indicating they are recombining with each other.

The fineStructure-linked coancestry model found the highest levels of shared genomic regions between isolates C4, C54 and C178 within clade B, providing evidence for strong haplotype sharing (Extended Data Fig. 5). These are all *MAT1-2* idiomorphs. Overall, however, strong haplotype donation occurred mostly within clades; an exception to this observation was the strong haplotype donation between C178, C4, C54 (clade B isolates) and C365 (a clade A isolate).

The average fixation index (F$_{ST}$) value was 0.1312 (range: 0–0.944035; s.d. = 0.0823); average F$_{ST}$ values and ranges for each chromosome are detailed in Supplementary Table 7. Chromosome 1 displayed extremely variable F$_{ST}$ values (range: 0–0.9440) (Fig. 3b). In particular, a region of 590 kilobase pairs (kbp) on the right arm of this chromosome displayed an average F$_{ST}$ value of 0.2273 but with a range in F$_{ST}$ values from 0 to 0.944035, suggesting near panmixis in some parts of this region between clades A and B. Across this region, 184 genes were found (gene ID Afu1g15860 to Afu1g17640 (Supplementary Data 4)). Of these genes, 9 contained SNPs that were significantly associated with itraconazole resistance using treeWAS. Also within this region, three extremely high

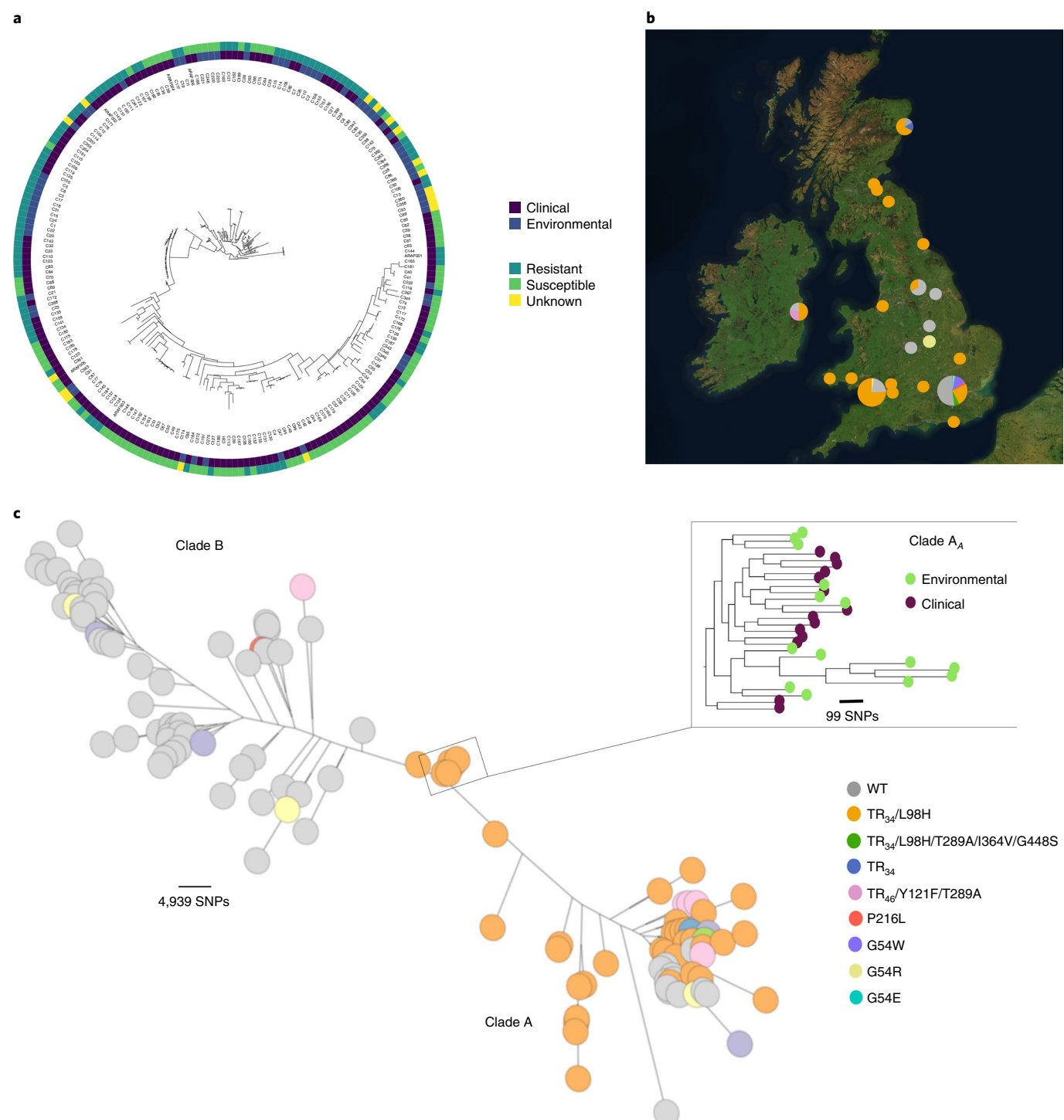

**Fig. 1 | Phylogeographical and phenotypic variation of 218 *A. fumigatus* isolates in the UK and Republic of Ireland. a**, Unrooted maximum likelihood phylogenetic tree (constructed in RAxML using genome-wide SNPs) showing the itraconazole MIC breakpoint (defined as above or below 2 mg l⁻¹ for resistance or susceptibility, respectively) and clinical or environmental source of isolation. **b**, Map showing the location of isolation for isolates included in this study, with the legend (bottom right) indicating the *cyp51A* polymorphisms present. **c**, Unrooted maximum likelihood phylogeny of all 218 isolates showing the genetic relationship between isolates. 'Clade A' and 'clade B' indicate the clustered nature of triazole resistance polymorphisms. The subclade in the midpoint of the phylogeny indicates a clonal clade, clade A_A, which is rich in clinical and environmental *A. fumigatus* isolates that contain the drug resistance polymorphism TR_34/L98H, highlighted in the inset phylogeny.

outlier $F_{ST}$ values were observed where the average $F_{ST}$ value was 0.9321 (range: 0.9238–0.9440) with the spanned regions containing 9 genes (Supplementary Table 8). Regions of higher than average $F_{ST}$ (approximately 0.5) were also observed in chromosomes 4 and 7

(Fig. 3b), implying population subdivision. The average $F_{ST}$ value for the *cyp51A* region, found in chromosome 4, was 0.1193.

Highly variable Tajima's *D* values were observed for both clades (Fig. 3a), suggestive of differing patterns of demography and

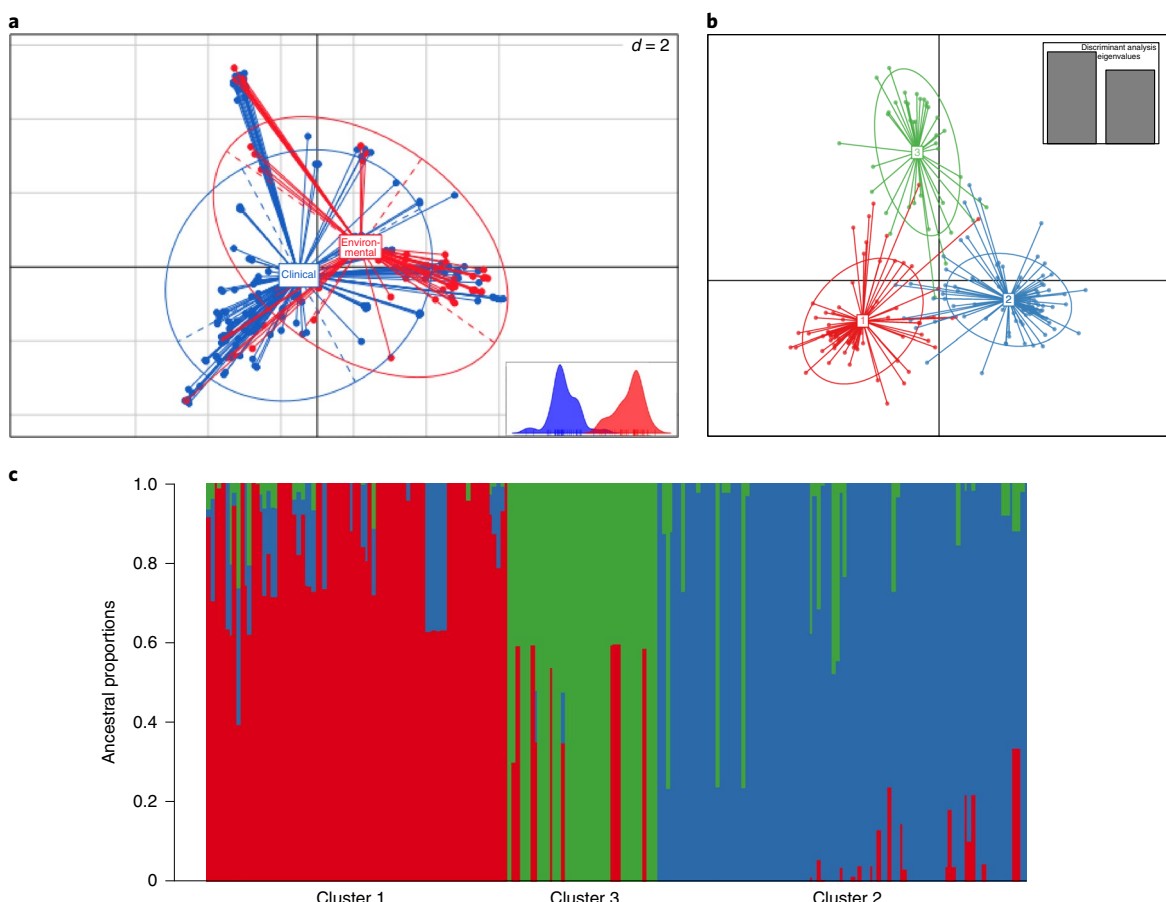

**Fig. 2 | Occurrence of three subclusters within the *A. fumigatus* population; clinical and environmental isolates were drawn from a single population.** **a**, Scatterplot of the PCA of *A. fumigatus* genotypes using the first two principal components illustrating genetic identity for clinical and environmental isolates. **b**, Discriminant PCA and PCA broadly identified three clusters, clusters 1–3, corresponding to the lowest BIC. **c**, Three subclusters were confirmed using STRUCTURE and $k = 3$.

natural selection. On average, Tajima's *D* was 0.4766 for clade A and −0.3839 for clade B (per the chromosome values in Supplementary Table 7). The region covering *cyp51A* appeared to have a lower than average *D* value (−1.02123; Fig. 3a) compared to the rest of clade A.

To determine the extent that variation in Tajima's *D* and $F_{ST}$ owe to intraclade population substructure, we subset the dataset into three clusters defined by PCA, discriminant PCA and STRUCTURE. Three-way $F_{ST}$ (Extended Data Fig. 5) between the three clusters showed regions in chromosome 1 approaching an $F_{ST}$ of 1, showing that these highly diverged alleles were present across clade A. Highly variable $F_{ST}$ values when comparing clusters 1 and 3 were observed (range: 0–0.786203; average: 0.1309), clusters 1 and 2 (range: 0–0.9322; average: 0.1577) and clusters 2 and 3 (range: 0–1; average: 0.1424). Estimating Tajima's *D* in clusters 1, 2 and 3 also found highly variable positive and negative values of *D* across all 3 clusters (Extended Data Fig. 6). The average value of *D* for all three clades was around zero, a marked departure from the previous analysis of solely clade A (cluster 1: −0.089152, range: −2.40694 to 4.68301; cluster 3: 0.01773, range: −2.62465 to 4.901) suggesting that the positive signature of selection in clade A was largely owed to population substructure.

**Gene–phenotype associations mirror regions of high $F_{ST}$.** tree-WAS is a microbe-specific approach that utilizes a phylogeny-aware approach to performing genome-wide association while being robust to the confounding effects of clonality and genetic structure. The algorithm identified 2,179 significant loci using the subsequent test; of these 1,385 significant loci were located within chromosome 4 and 64% of significant loci ($n = 1,391$) were found to be intergenic. The 3 most significant loci ($P < 2.03 \times 10^{-6}$) were all located on chromosome 4: 1 was intergenic (position 1,779,747), one was a non-synonymous SNP (S237F) in Afu4g07010 (position 1,816,171) and the third was the L98H substitution in *cyp51A* (position 1,784,968; Supplementary Data 1). Other significant loci mapped to genes involved in secondary metabolism, including fumitremorgin. There were also loci within Afu8g00230, encoding verruculogen synthase, which is associated with *A. fumigatus* hyphae and conidia modifying the properties of human nasal epithelial cells[26]. A single significant SNP was also identified in *Aspf2* (Afu4g09580), which is involved in immune evasion and cell damage[27].

Peaks of gene–phenotype associations on chromosomes 1, 4 and 7 mirrored regions of high $F_{ST}$ when comparing clades A and B (Fig. 3b) probably reflecting the impact of azole selection on these alleles. An exception to this was observed in chromosome 8, where $F_{ST}$ values did not peak above 0.370951 but treeWAS *P* values were significant. Significant loci within chromosome 8 are located in genes such as verruculogen synthase, polyketide synthase (PKS)-non-ribosomal peptide synthetase (NRPS) hybrid synthetase *psoA* (Afu8g00540) and brevianamide F prenyltransferase (Afu8g00210). Out of the 356 significant loci in chromosome 8, most ($n = 354$) were located within 500,000 bp of the start of chromosome; 62% were intergenic.

To determine the link between significant loci identified by treeWAS and azole resistance, preliminary investigations were

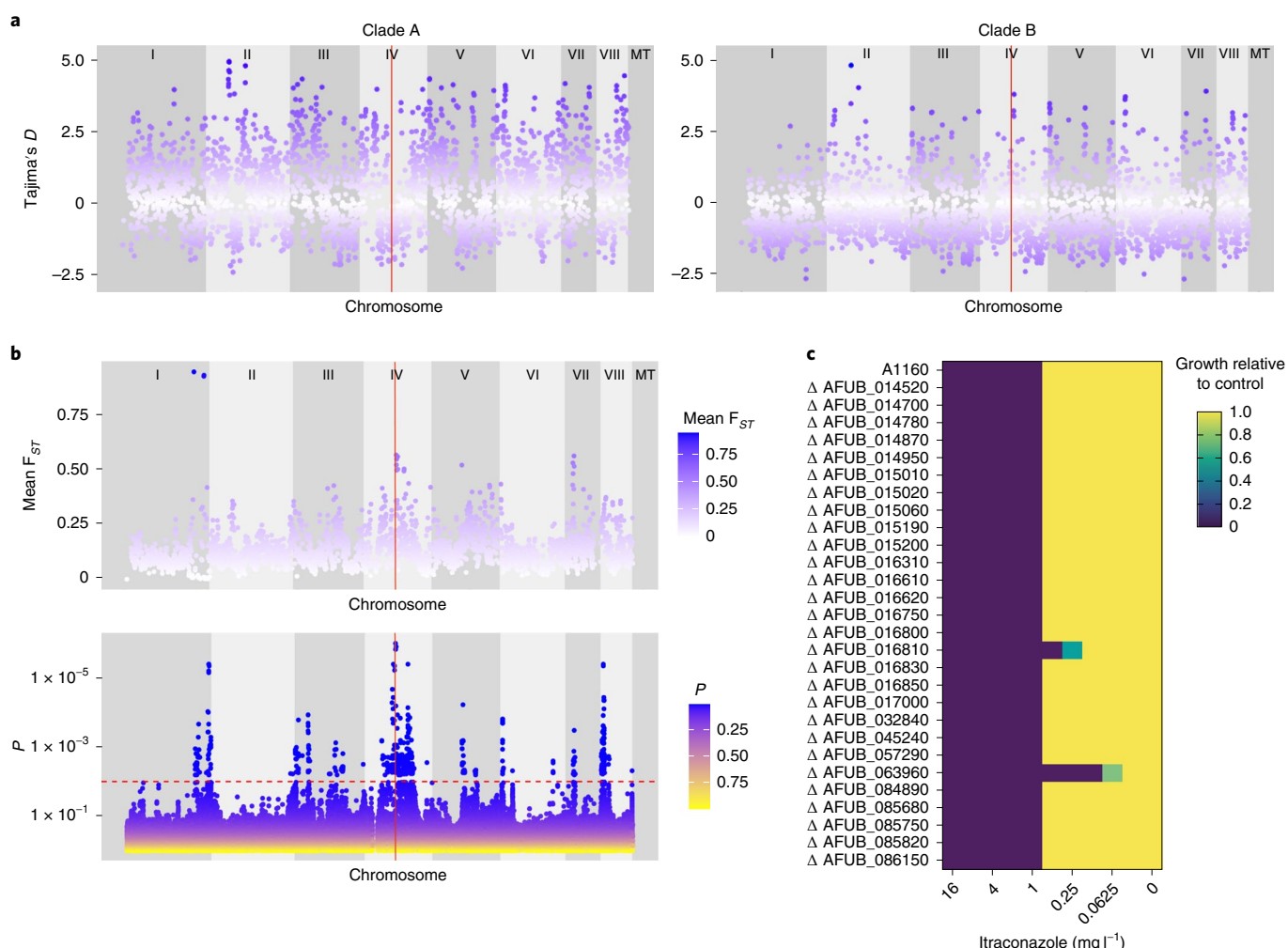

**Fig. 3 | Loci associated with itraconazole resistance linked to regions of high F$_{ST}$ and selection. a**, Scatterplot of Tajima's $D$ estimates for each chromosome for all isolates within clade A (left) and scatterplot of Tajima's $D$ estimates for each chromosome for all isolates within clade B (right). The position of *cyp51A* is highlighted in red. **b**, Scatterplot of sliding 10-kb non-overlapping window estimates of F$_{ST}$ for each chromosome between isolates within clades A and B (top). Manhattan plot (bottom) for the treeWAS subsequent test (bottom) showing $P$ values for all loci and a significant threshold of 0.01 (dashed red line), above which points indicate significant associations. The vertical red line in both plots denotes the position of *cyp51A*. **c**, Relative growth of null mutants (compared to A1160) of genes with significant loci identified in treeWAS on media containing itraconazole.

undertaken using 28 gene deletion mutants from the COFUN knockout collection where the corresponding loci were deleted. As expected, the null mutant ΔAFUB_063960 (*cyp51A*) was not able to fully grow in medium containing >0.06 mg l⁻¹ itraconazole. The null mutant ΔAFUB_016810 (*abcA*) was unable to grow in medium containing >0.25 mg l⁻¹ itraconazole (Fig. 3c). The control strain and all other null mutants were unable to grow in medium containing >1 mg l⁻¹ (Fig. 3c and Table 1). Potentially, these genes may work in combination with others to confer drug resistance.

**Clinical and environmental isolates are highly related.** PCA and discriminant PCA showed a lack of genetic differentiation among clinical and environmental isolates, showing that clinical isolates are drawn from a wider environmental population (Fig. 2a). Phylogenetically, 6 pairs or groups of *A. fumigatus* contained both clinical and environmental isolates that were genetically very highly related (Supplementary Table 9), with bootstrap support of 65% or higher (range: 65–100%: median = 100%), from both clades A and B. For these pairs or groups, the average pairwise diversity was 297 SNPs (2.5% of total diversity); 4 contained

TR$_{34}$/L98H. On average, any clinical/environmental pair or group of azole-resistant isolates were separated by 247 SNPs (range: 227–270 SNPs).

The pair or group showing the highest identity was group 5 in clade B, containing the clinical isolates C42, 43, 44 and 48 and the environmental isolate C96. These isolates were separated by only 217 SNPs (1.8% of the total diversity seen in this dataset), contained the *MAT1-2* idiomorph and did not contain any polymorphisms within *cyp51A* associated with drug resistance nor had raised MICs (Supplementary Table 2).

An additional cluster of very highly related *A. fumigatus* isolates consisting of 14 clinical and 14 environmental isolates all harboured the TR$_{34}$/L98H *cyp51A* polymorphism and *MAT1-2* idiomorph. This clonal clade sits within the larger clade A and within cluster 3 and will henceforth be referred to as clade A$_A$ (Fig. 1c). Isolates within clade A$_A$ were found to be broadly distributed across England, Wales, Scotland and Ireland, covering a spatial distance equivalent to the whole dataset (Fig. 1b). Clade A$_A$ appeared with high frequency, comprising 13% of the total dataset and 23% of the isolates found within clade A. The average number of SNPs separating the clade A$_A$ isolates was 294 SNPs (2.48% of total genetic

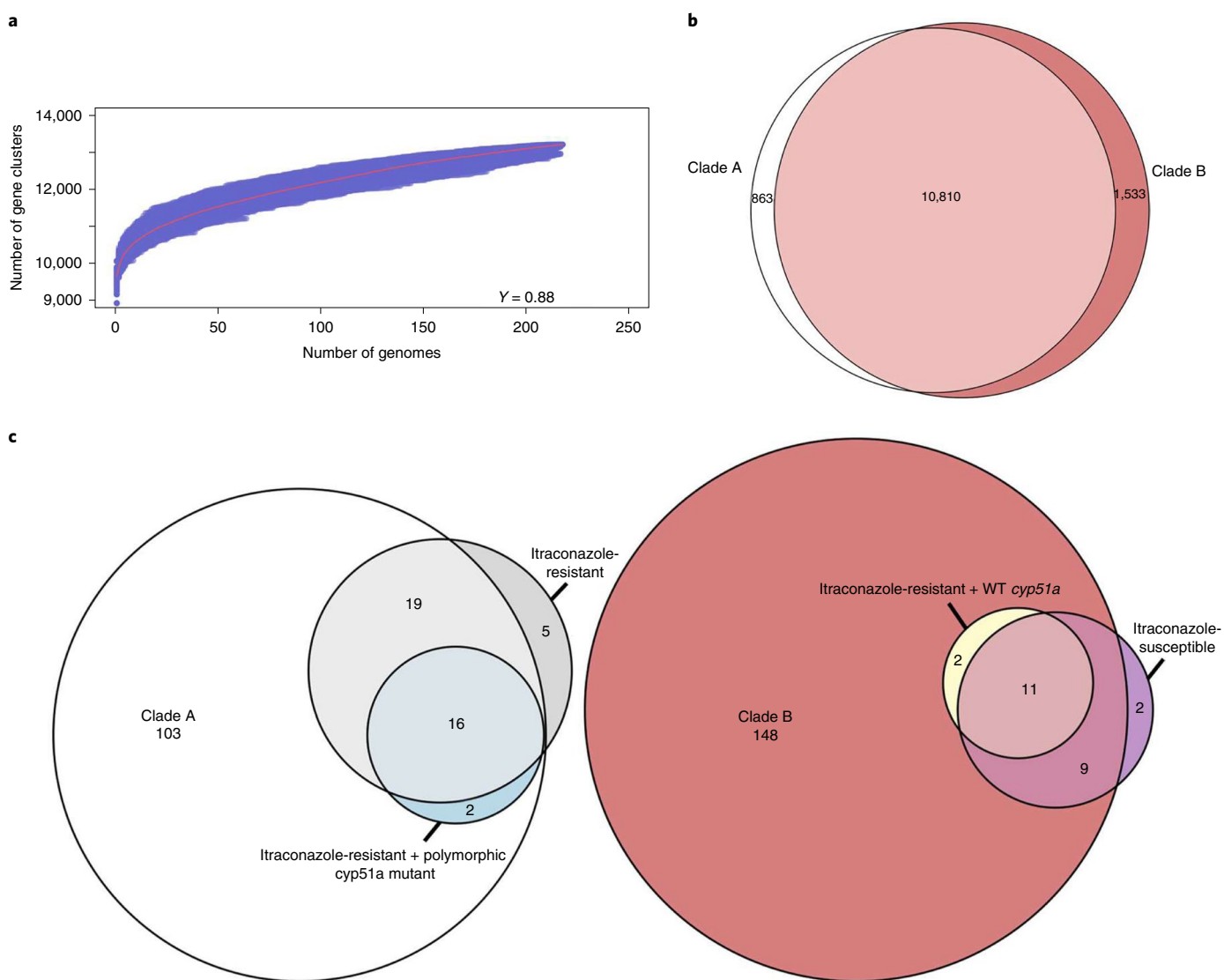

**Fig. 4 | Statistical analysis of the open pan-genome associates genes from itraconazole-resistant isolates within either clade A or B based on the prevalence of WT *cyp51a*. a**, Plot of Tettelin's model on the pan-genome generated from all isolates. The red line indicates the median value from all permutations and the $\gamma$ value indicates that the pan-genome ($\gamma > 0$) is classed as 'open'. **b**, Venn diagram plotting genes occurring at least once within isolates from clades A and B and genes only found within isolates from clade A or B. **c**, Venn diagram plotting genes associated with clades A and B, itraconazole resistance/susceptibility and WT/mutant *cyp51a*. Associated genes obtained a Bonferroni $P < 0.05$ and were organized into groups based on log ORs, from the Scoary results. The numbers represent the gene numbers.

diversity) and had high bootstrap support (100%) within the phylogeny (Extended Data Fig. 2). Isolation by distance showed no significant correlation (observation = 0.0379) between genetic and geographical distances suggesting that the clade $A_A$ clone will have a broader extent than just the British Isles.

Clade $A_A$ was significantly overrepresented (chi-squared test $P = 2.75 \times 10^{-13}$, d.f. = 1) for drug-resistant environmental isolates, compared to only 63% of environmental isolates in the rest of the dataset. In comparison, only 54% of clinical isolates within the dataset contained the TR$_{34}$/L98H polymorphism. Two pairs of isolates from clinical and environmental sources in clade $A_A$ showed high genetic identity between clinical and environmental isolates—pairs 1 and 2 (Supplementary Table 9). Pair 1 showed the highest identity, with 227 SNPs separating environmental isolate C21 and clinical isolate C112. Nucleotide diversity ($\pi$) tests showed that the genetic diversity separating pair 1 isolates was significantly different to the mean nucleotide diversity within the clade $A_A$ isolate (one-tailed $t$-test $P < 3.0473 \times 10^{-252}$; Supplementary Table 9), showing that

pair 1 isolates were genealogically tightly linked and that this shared identity could not have occurred by chance occurrence alone.

**Accessory gene content is associated with drug resistance.** By utilizing the WGS performed in this study, we generated an *A. fumigatus* pan-genome of 218 isolates. De novo assemblies produced genome sizes between 27.3 and 31 Mb (average = 28.8 Mb) and had high contiguity, with a mean N50 of 12,843 bp and mean number of contigs of 5,652 (range: 5,039–8,754).

The pan-genome was validated against 55 essential *Af*293 genes[28]; 53 of these were found in >99% of isolates. The rest were found in 156 (AFUA_2G09060) and 173 (AFUA_4G11280) isolates but all were discovered (Supplementary Table 10). In total, our pan-genome identified 13,206 genes, comprising 6,705 core genes and 6,501 accessory genes. Approximately 50% of genes were shared between all isolates and 1,592 genes were isolate-specific (approximately 12% of the pan-genome) with an average of 7 new genes per isolate. Tettelin's model[29] showed that the pan-genome is classed as 'open' ($\gamma > 0$),

**Table 1 | MIC that null mutants of genes identified as being statistically significant loci by treeWAS stop growing at, against the A1160 control**

| A1160 gene ID | Af293 gene ID | MIC (mg l⁻¹) | treeWAS P | Gene description | Locus |
|---|---|---|---|---|---|
| AFUB_063960 | AFUA_4G06890 | >0.0625 | $<2.03 \times 10^{-6}$ | *cyp51A* | L98H |
| AFUB_016810 | AFUA_1G17440 | >0.25 | 0.001546976 | *abcA* | Y1149N |
| AFUB_014520 | AFUA_1G14960 | >1 | 0.001546976 | Has domain(s) with predicted actin binding activity | A820V |
| AFUB_014700 | AFUA_1G15150 | >1 | 0.003190639 | Orthologue(s) have hydrolase activity | L100F |
| AFUB_014780 | AFUA_1G15230 | >1 | 0.004462783 | Orthologue(s) have role in conidiophore development | Synonymous SNP |
| AFUB_014950 | AFUA_1G15410 | >1 | 0.000502209 | Has domain(s) with predicted zinc ion binding activity | S109P |
| AFUB_015010 | AFUA_1G15460 | >1 | 0.000200224 | Orthologue of *Aspergillus nidulans* FGSC A4: AN0892 | Synonymous SNP |
| AFUB_015020 | AFUA_1G15470 | >1 | 0.005395385 | Orthologue(s) have RNA polymerase II transcription factor activity | Synonymous SNP |
| AFUB_015060 | AFUA_1G15520 | >1 | 0.001956814 | Orthologue(s) have allophanate hydrolase activity | F2S |
| AFUB_015190 | AFUA_1G15660 | >1 | 0.005395385 | Orthologue of NRRL 181: NFIA_009670 | L71S |
| AFUB_015200 | AFUA_1G15670 | >1 | 0.001956814 | Putative laccase | Synonymous SNP |
| AFUB_016310 | AFUA_1G16920 | >1 | 0.005567187 | Beta-xylosidase | Synonymous SNP |
| AFUB_016610 | AFUA_1G17220 | >1 | 0.0000936 | Putative secreted polygalacturonase GH-28 | Intron |
| AFUB_016620 | AFUA_1G17230 | >1 | 0.000359083 | Orthologue(s) have carbon-oxygen lyase activity | P466T |
| AFUB_016750 | AFUA_1G17380 | >1 | 0.000200224 | Has domain(s) with predicted oxidoreductase activity | I68V |
| AFUB_016800 | AFUA_1G17430 | >1 | 0.00000228 | Orthologue(s) have monophenol monooxygenase activity | E586D |
| AFUB_016830 | AFUA_1G17470 | >1 | 0.00000305 | *nrtB* | Intron |
| AFUB_016850 | AFUA_1G17490 | >1 | 0.001152366 | Has domain(s) with predicted carbohydrate binding | G298S |
| AFUB_017000 | AFUA_1G17620 | >1 | 0.000887684 | Orthologue of RIB40: AO90011000197 | Synonymous SNP |
| AFUB_032840 | AFUA_2G17190 | >1 | 0.005886174 | Has domain(s) with predicted ATP binding | P1013A |
| AFUB_045240 | AFUA_3G03010 | >1 | 0.000887684 | Putative phosphate-repressible phosphate permease | Synonymous SNP |
| AFUB_057290 | AFUA_5G09740 | >1 | 0.004617329 | Orthologue(s) have role in conidiophore development | E432K |
| AFUB_084890 | AFUA_8G01700 | >1 | 0.003128212 | Has domain(s) with predicted catalytic activity | Synonymous SNP |
| AFUB_085680 | AFUA_8G00900 | >1 | 0.002175055 | Orthologue of FGSC A4: AN8368 | T47M |
| AFUB_085750 | AFUA_8G00820 | >1 | 0.000397402 | Orthologue of RIB40: AO90138000119 | H16R |
| AFUB_085820 | AFUA_8G00750 | >1 | 0.0000406 | Has domain(s) with predicted DNA binding | Synonymous SNP |
| AFUB_086150 | AFUA_8G00420 | >1 | 0.001260979 | C6 finger transcription factor *fumR* | N305D |

indicating that the total number of genes within the pan-genome had not reached saturation for the 218 genomes (Fig. 4a).

A total of 10,810 genes were found at least once in isolates from clades A and B, whereas 863 and 1,533 genes were only present in isolates from clades A and B, respectively (Fig. 4b); 140 genes were statistically associated with clade A and 170 with clade B (Supplementary Data 5). Thirteen genes were found within clade B itraconazole-resistant isolates with WT *cyp51A*, indicating the presence of clade-specific genes (Supplementary Data 6). In contrast, clade A included 37 genes that also associated themselves with either itraconazole resistance (*n* = 19), itraconazole resistance with *cyp51A*

polymorphism (*n* = 2) or both (*n* = 16) (Supplementary Data 7). No genes affiliated with itraconazole-susceptible isolates were found within clade A (Fig. 4c). This finding indicates that enriched genes in clade A were strongly associated with polymorphisms in *cyp51A* compared to the gene content of clade B where enriched genes were associated with alternative unmapped resistance mechanisms.

## Discussion

Breakthrough infections by azole-resistant *A. fumigatus* have been observed with striking increases across northern Europe, where the incidence has increased from negligible levels pre-1999 up to a

measured prevalence of 3–40% in the present day[11,30]. Furthermore, the advent of COVID-19 has created a large and growing global cohort of patients that are at risk of azole-resistant *A. fumigatus* coinfections[3]. In this study, we completed WGS of 218 clinical and environmental *A. fumigatus* isolates to explore the epidemiology of azole-resistant aspergillosis.

Our genomic analysis of environmentally and clinically sourced *A. fumigatus* yielded four main findings. First, we identified strong genetic structuring into two clades A and B, with most environmentally occurring azole resistance alleles segregating inside clade A and showing signatures of selection at multiple loci, some of which are known to adapt in response to selection by fungicides. Phylogenomic analysis confirmed that the population of *A. fumigatus* was not panmictic; it is structured into a characteristic 'dumb-bell' phylogeny marked by two clades with limited interclade recombination. We used four hypothesis-free population genetic methods, PCA, discriminant PCA, STRUCTURE and fineStructure to independently confirm the existence of the two clades, as well as the occurrence of strong genetic subclustering within clade A and evidence of widely occurring clonal genotypes (for example, clade A$_A$). Clade A contained most isolates (88%) harbouring polymorphisms in *cyp51A* and associated with drug resistance compared to clade B, which was predominantly WT for *cyp51A*. Phenotypically defined resistance recovered a similar pattern with 78% of itraconazole MICs above clinical breakpoints being compartmentalized into clade A. Further analysis using PCA, discriminant PCA and STRUCTURE showed that clade A is divided into 2 subclusters, clusters 1 and 3. All TR$_{34}$/L98H polymorphisms were found within cluster 3 while cluster 1 contained isolates with a variety of *cyp51A* polymorphisms. Interestingly, isolates containing the TR$_{46}$/Y121F/T289A polymorphism were found in both clade B and clade A cluster 1 but not clade A cluster 3. These analyses suggest that TR-associated azole resistance has evolved a limited number of times and recombination has not yet had the impact of homogenizing these alleles across the wider *A. fumigatus* phylogeny. Similar conclusions were drawn by Camps et al.[31] based on microsatellite and cell surface protein marker analysis of European isolates, who also suggested that the TR resistance form had developed from a common ancestor or restricted set of genetically related isolates. Inclusion of non-UK publicly available data confirmed the two-clade structure, with most environmentally associated drug resistance alleles occurring within clade A. Additionally, other non-UK studies have assigned isolates to clades A and B, further confirming the two-clade structure globally[24,32]. However, future work on global collections of *A. fumigatus* that have been collected across longer periods of time than in the current study will be needed to trace and date the spatiotemporal origins of these alleles.

fineStructure enabled the confirmation of three subpopulations within this dataset, which corresponded to clades A, B and A$_A$, and the presence of a subtle population substructure, which has been described in other studies[33]. Strong donation of haplotypes between the clade B isolates C4, C54 and C178 and the clade A isolate C365 is indicative of recent recombination between these two clades, highlighting the potential for recombination-driven introgression of resistance alleles to occur. Previous studies have confirmed that asexual reproduction facilitates the emergence of TR$_{34}$/L98H within *A. fumigatus*[34]; this mechanism may also facilitate the emergence of new resistance alleles across the population. However, the relative impact of sexual/asexual recombination as mechanisms responsible for facilitating the emergence and/or spread of resistance alleles is to be determined. The lack of a reliable molecular clock for *A. fumigatus*, combined with evidence of recombination between the clades, currently hinders the dating of the time of emergence of resistance alleles.

Second, we observed multiple exemplars of drug-resistant genotypes in the clinic that matched those in the environment with very high identity. Since patients have never convincingly been shown to transmit their *A. fumigatus* to the environment, this finding demonstrates that at-risk patients were infected by isolates that have pre-acquired their resistance to azoles in the environment. The statistically significant association between the azole-resistant isolates C354, C355 and C357, two clinical isolates and an environmental isolate, respectively, isolated within the same city, coupled with their very high genetic similarity suggests an environment-to-patient acquisition of azole-resistant *A. fumigatus* with very high confidence. The low number of SNPs separating the isolates and nucleotide diversity tests showed that these isolates are genealogically tightly linked and were strongly supported phylogenetically with 100% bootstrap support. This shared identity was statistically significant ($P < 3.0473 \times 10^{-252}$) and could not have occurred by chance alone. Our PCA analysis showed that clinical isolates were drawn from a wider environmental diversity, with a lack of genetic differentiation among isolates in these populations (Fig. 2a). The presence of other significant shared identity between clinical and environmental isolates (Supplementary Table 8) demonstrates that this environmental-to-clinical acquisition via inhalation of fungal spores is not a rare occurrence.

Third, gene–phenotype associations mirrored regions of high $F_{ST}$, suggesting that these itraconazole-resistant phenotypes are linked not only to regions of the genome under selection but also to the genetic structure that we observed. We found a striking congruence where significant peaks of gene–phenotype associations on chromosomes 1, 4 and 7I mirrored regions of high $F_{ST}$ when compared to clades A and B (Fig. 3a). It appears that fungicide-associated resistance is driving the patterns of evolution and is the most likely explanation for much of the observed genetic architecture. treeWAS also identified significant SNPs in genes involved in secondary metabolism. Recent research has shown that secondary metabolites combat the host immune system and aid growth in the host (human) environment[35]. Our results identified four SNPs in the gene encoding fumitremorgin C monooxygenase (Afu8g00240), which are part of the pathway for secondary metabolite fumitremorgin C, a mycotoxin that acts as a potent ATP-binding cassette sub-family G member 2/breast cancer resistance protein inhibitor that reverses multidrug resistance[36]. This pathway has also been suggested as regulating the brevianamide F gene cluster[37]; two non-synonymous SNPs in the gene encoding brevianamide F prenyltransferase (Afu8g00210) were also identified as significant. Further complexity was observed in chromosome 8, where $F_{ST}$ values did not peak above 0.370951 but treeWAS $P$ values were highly significant. Significant loci within chromosome 8 were located in genes such as verruculogen synthase, PKS-NRPS synthetase *psoA* (Afu8g00540) and brevianamide F prenyltransferase (Afu8g00210). Future work now needs to widen our search and focus on the use of reverse functional genomic approaches to interrogate the function of these genes within the context of their potentially epistatic interrelationships with the canonical ergosterol biosynthesis *cyp51A* azole resistance alleles on chromosome 4.

Finally, through pan-genome analysis, we showed that *A. fumigatus* has an extensive, variable accessory genome that expands with the addition of each new genome (Fig. 4a). These results are in contrast to previous studies[38,39], possibly due to differing numbers of isolates occupying different geographies and habitats, as well as alternative bioinformatics pipelines[38,39]. From the analysis of the accessory gene content, we found that genes linked to itraconazole resistance and/or known resistance-associated *cyp51a* polymorphisms were found strictly within clade A (Fig. 4c). Within clade B, we found evidence for genes in itraconazole-resistant isolates that contained no known *cyp51a* drug resistance polymorphisms. However, a large proportion of these genes were also found in itraconazole-susceptible isolates (11 out of 13, 85%) (Fig. 4c). These findings point to the existence of new resistance mechanisms with

an underlying polygenic basis and show that further understanding the role of accessory genes in generating drug-resistant phenotypes is needed. Such studies may also shed light into the extent that resistance in *A. fumigatus* may be influenced by additional factors, such as the epigenetic-mediated resistance observed in *Mucor circinelloides*[40].

Therefore, this study supports the hypothesis that the widespread use of azole fungicides in agriculture is coupled to widespread isolation of azole-resistant *A. fumigatus* from environmental sources[14]. These isolates, in turn, bear hallmark multilocus genotypes that are indistinguishable to those recovered from patients, supporting our conclusion that adaptation to fungicides in the environment leads to transmission of *A. fumigatus*-bearing azole resistance genotypes without obvious detriment to their clinical fitness[41,42]. The identification of spatially widespread *A. fumigatus* clones that are not only resistant to azoles but are also highly represented in both the environment and clinic suggests that there are few fitness costs associated with this phenotype. Respiratory viruses such as H1N1 influenza are known to predispose critically ill patients to secondary mould infections[2] and case studies increasingly show that similar infections are experienced by patients with COVID-19 (ref. [3]). Therefore, the growing numbers of susceptible individuals underscores the need to more fully understand the risk posed by environmental reservoirs of pathogenic fungi that, primarily through the use of agricultural antifungals, have evolved resistance to first-line clinical azoles.

## Methods

**Fungal isolates.** A total of 218 isolates were included in this study, spanning 12 years (2005–2017), covering a spatial range of 63,497 miles[2] in England, Wales, Scotland and Ireland (Fig. 1b); 153 clinical *A. fumigatus* isolates from 5 participating centres were included. Patients either had respiratory colonization with *A. fumigatus* or were suffering from different manifestations of aspergillosis and had the following underlying disorders (Supplementary Table 1): CF (64%); other conditions (11%); unknown (25%). Environmental isolates (*n* = 65) were collected from the following sources: soil (72%); plant bulbs (12%); air (3%); compost (2%); and unknown (11%). Briefly, plant bulbs were purchased from a garden centre in Dublin, Ireland and swabbed and plated onto Sabouraud dextrose agar[22]. Soil samples were collected from 16 sites across south England between May and July 2018; locations were selected to incorporate a range of habitat types. Dry soil was loosened and collected into a 5 ml Eppendorf tube (Eppendorf AG)[43]. Soil and air samples from south Wales were collected from June to November 2015 in urban and rural locations[44].

Many isolates from both clinical and environmental sources were specifically selected for WGS because they displayed phenotypic azole resistance (raised MICs to at least one triazole drug using EUCAST or CLSI) and did not constitute a randomized sample. We report MICs for itraconazole (*n* = 200), voriconazole (*n* = 201) and posaconazole (*n* = 177) in Supplementary Table 7.

All referred isolates were identified by phenotypic morphology because the *A. fumigatus* species complex was based on colonial morphology and microscopic characteristics. Isolates were cultured on Sabouraud dextrose agar (Oxoid) and malt extract agar (Sigma-Aldrich) at 37 °C ± 2 °C for 5–7 d in the dark. The adhesive tape technique was used for microscopic examination of fungal cultures. Slides were prepared with lactophenol cotton blue as the mounting and staining fluid. The Atlas of Clinical Fungi (https://www.clinicalfungi.org) was consulted as an identification reference. In addition, growth at 45 °C was used to exclude most cryptic species within section *Fumigati*. Isolates with elevated azole MICs were confirmed to be *A. fumigatus* by matrix-assisted laser desorption ionization–time of flight mass spectrometry, performed with a Microflex LT system (Bruker) using the Biotyper 3.0 software with the additional fungi library (Bruker) according to the manufacturer's recommendations or as described by Dunne et al.[22]. Exact identification of azole-resistant *A. fumigatus* isolates from two participating centres in London, UK, was confirmed by sequencing the partial calmodulin gene (*CaM* locus) as described previously[45]. Antifungal susceptibility testing was completed as part of the original sampling studies or determined according to the standard EUCAST[46] or CLSI M38-A2 broth microdilution methods[47]. MICs for itraconazole and voriconazole were determined for 92% of isolates (*n* = 200 and 201, respectively). MICs for posaconazole were determined for 81% of isolates (*n* = 177). Seventeen isolates (8%) were not tested for susceptibility and therefore have no recorded MICs for any antifungal drug.

**DNA preparation and WGS.** High-molecular-weight DNA was extracted from all 218 isolates and quantified. Briefly, high-molecular-weight DNA was extracted using the MasterPure Yeast DNA Purification Kit (Epicentre Biotechnologies) with bead beating with 1.0 mm zirconia/silica beads (BioSpec Products) in a FastPrep-24 system (MP Biomedicals) at 4.5 m s⁻¹ for 45 s. Genomic DNA was quantified with a Qubit 2.0 fluorometer and dsDNA BR Assay Kit (Thermo Fisher Scientific) and quality-controlled with a TapeStation 2200 (Agilent Technologies) and gDNA ScreenTape assays (Agilent Technologies). Genomic DNA libraries were constructed with the Illumina TruSeq Nano Kit (Illumina) at the Natural Environmental Research Council (NERC) Biomolecular Analysis Facility (BAF), University of Edinburgh, UK (http://genomics.ed.ac.uk/). Prepared whole-genome libraries were sequenced on an Illumina HiSeq 2500 sequencer at BAF, generating 150 bp paired-end reads in high output mode.

**Bioinformatic analysis.** All raw Illumina paired-end reads were quality-checked with FastQC v.0.11.5 (Babraham Institute) and aligned to the reference genome *Af*293 (ref. [48]) using the Burrows–Wheeler Aligner v.0.7.8 (ref. [49]) MEM and converted to sorted BAM format using SAMtools v.1.3.1 (ref. [50]). Variant calling was performed with GATK[51,52] HaplotypeCaller v.4.0, excluding repetitive regions identified using RepeatMaster[53] v.4.0.6. Low-confidence variants were filtered out providing they met at least 1 of the parameters DP < 10 || RMSMappingQuality < 40.0 || QualByDepth < 2.0 || FisherStrand > 60.0 || ABHom < 0.9. All variant calls with a minimum genotype quality of less than 50 were also removed using a custom script. SNPs were mapped to genes using vcf-annotator (Broad Institute). TRs and SNPs causing non-synonymous amino acid substitutions in *cyp51A* encoding 14-α lanesterol demethylase, the target of triazole antifungals, are summarized in Supplementary Table 1 for all isolates where present. Identified polymorphisms were compared to known polymorphisms conferring resistance using the MARDy database v1.1[54]. Changes in ploidy were analysed by interrogating the normalized depth of coverage per isolate using bamCoverage from the deepTools 2.0 package[55]. Mating type was reported for all isolates: approximately a third of isolates (34%; *n* = 79) contained the *MAT1-1* mating type idiomorph (Supplementary Table 2).

**Phylogenetic analysis.** Whole-genome SNP data were converted into presence/absence of an SNP with regard to the reference. SNPs identified as low confidence in the variant filtration step were converted to missing data. These data were converted into relaxed interleaved Phylip format and maximum-likelihood phylogenies were constructed to assess sequence similarity between isolates using rapid bootstrap analysis over 1,000 replicates using the GTRCAT model of rate heterogeneity in RAxML[56] v.8.2.9. Phylogenies were visualised in FigTree v.1.4.2.

**Analysis of genetic diversity and population inference.** Isolation by distance was examined via a Mantel test implemented in adegenet[57] in R v.3.5.3 with a simulated *P* of 0.01.

Previous studies used nucleotide diversity (*π*) as a metric to test whether pairs of isolates are epidemiologically linked to infer transmission[58,59]. Nucleotide diversity (*π*) tests were implemented in VCFtools[60] v.0.1.13.

Genetic similarity and population allocation was investigated via hypothesis-free approaches. PCA and discriminant PCA[61] were performed to interrogate the relationship between clinical and environmental isolates based on SNP data using the R package adegenet[57] v.2.1.1. Genetic clusters were allocated based on identifying the optimal number of clusters (*k*) corresponding to the lowest Bayesian information criterion (BIC). STRUCTURE v.2.3.4 (ref. [62]) was used to independently investigate the population structure assignments that were predicted by discriminant PCA and PCA.

We analysed a coancestry matrix based on whole-genome SNP data to assign individuals to populations via model-based clustering using fineStructure[63] v.2.0.7. fineStructure uses chromosome painting to identify important haplotypes and describe shared ancestry within a recombining population. These analyses were performed using a pan-clade genome-wide SNP matrix to infer recombination, population structure and assignment, and the ancestral relationships of all lineages. ChromoPainter reduced the SNP matrix to a pairwise similarity matrix under a linked model, using information on linkage decay and reducing the within-population variance of the coancestry matrix relative to the between-population variance.

Sliding window population genetic statistics (Tajima's *D*, nucleotide diversity (*π*) and $F_{ST}$) were calculated for non-overlapping windows of 10 kb using VCFtools[60] v.0.1.13, with resulting graphics produced in R v.3.5.3 using ggplot2. We measured the signatures of genome-wide population differentiation via the $F_{ST}$ analysis of non-overlapping 10 kb windows by comparing isolates within clade A against those isolates from clade B and for each cluster identified by PCA, discriminant PCA and STRUCTURE; the $F_{ST}$ is a measure of population differentiation due to genetic structure, with values ranging from 0 (implying complete panmixis) to 1 (no genetic diversity is shared between the two populations). Tajima's *D*[64] measures departures from neutral expectations and selection.

The index of association and rBarD are commonly used to estimate linkage disequilibrium. These two statistics were calculated using Poppr v.2.8.5 (ref. [65]) in R v.3.5.3 using 999 resamplings of the data under the null hypothesis of no linkage disequilibrium and that no recombination would not be rejected.

**Identifying loci associated with itraconazole resistance.** Loci associated with itraconazole resistance were identified using treeWAS[66], a method recently developed to address challenges specific to microbial GWAS. treeWAS uses phylogenetic information to correct for the microbial population structure; therefore, we used the phylogeny presented in this study along with a nucleotide alignment for all 218 isolates. treeWAS was performed for all isolates with itraconazole MIC information and a binary phenotype, categorized as itraconazole MIC defining susceptibility as MIC $< 2\,\mathrm{mg\,l^{-1}}$ and resistant as MIC $\geq 2\,\mathrm{mg\,l^{-1}}$ ($n = 200$) with a $P$ cut-off of 0.01 for 3 tests of association (subsequent, simultaneous and terminal) between azole susceptibility phenotype and genotype. However, only the subsequent test results were used because this test was previously defined by Collin and Didelot[66] as most effective at detecting subtle patterns of association. A phylogeny for these isolates was reconstructed as previously described with 281,874 SNPs common to 1 or more isolates with reported itraconazole MIC. The phenotype information was encoded as a discrete vector based on above the MIC breakpoint for itraconazole (and therefore resistant) or below (susceptible). Significant SNPs common to all three tests (Supplementary Data 1–3) were combined and were mapped to their respective genes via FungiDB release 50 beta[67].

**Drug sensitivity screening.** Null mutants were obtained from the COFUN genome-wide knockout collection[68] and the COFUN transcription factor knockout library[69]. MFIG001 (A1160p+) was used as the parental isolate[70]. Strains were inoculated in 25 cm² tissue culture flasks containing Sabouraud dextrose agar + 100 μM hygromycin and cultured for 3 d at 37 °C. Conidia were collected in PBS + 0.01% Tween-20 (Sigma-Aldrich) by filtration through Miracloth (VWR). Spore concentrations were determined by haemocytometer. Spores were inoculated in a CytoOne 96-well plate (StarLab) containing Roswell Park Memorial Institute 1640 medium 2% glucose and 165 mM 3-(N-morpholino)propanesulfonic acid buffer (pH 7.0) with 16–0.06 mg l⁻¹ itraconazole. MICs were determined visually after 48 h as outlined by EUCAST[71]. Each strain was assessed in biological triplicate relative to the parental control (strain A1160). Heatmaps were generated in Prism 9 (GraphPad Software).

**Pan-genome generation and accessory gene content analysis.** De novo assembled genomes were produced for all 218 isolates. First, sequencing data underwent quality control, trimming and adaptor removal using Trimmomatic[72] v.0.39 (ILLUMINACLIP:TruSeq3-PE.fa:2:40:15 TOPHRED33 LEADING:20 TRAILING:20 SLIDINGWINDOW:2:20 MINLEN:25 AVGQUAL:20). Cleaned sequencing reads were assembled using MEGAHIT[73] v.1.2.9 (--no-mercy). Low-complexity regions for each genome were masked using RepeatMasker[53] v.4.1.2-p1 and the Dfam repeat database release 3.5[74], to improve the speed of downstream gene calling. The RepeatMasker parameters were optimized for *A. fumigatus* (-species '*Aspergillus fumigatus*' -s -no_is -cutoff 255 -frag 20000). Additionally, assembly statistics were generated using BBTools[75] v.37.62. GeneMark-EP and ProtHint[76] were combined into a semi-supervised gene calling pipeline that utilized a set of know proteins to predict potential orthologues within a genome. Orthologous proteins were obtained from OrthoDB[77] v.10.1 and were used to train ProtHint v.2.6.0; GeneMark-EP v.4.68 was used to predict genes on isolate genomes. Assembly statistics optimized the minimum contig size for prediction; assemblies whereby contigs >5 kb covered <40% of the genome had a minimum contig size set to 5 kb; otherwise, the minimum contig size was 1 kb (--max_intro 200 --max_intergenic 100000 --min_contig=1000 --min_contig_in_predict = (5000/1000)). Proteins were subsequently extracted using the GeneMark-EP built-in script.

To uncover commonly shared genes and enable pan-genome construction, a protein BLAST[78] v.2.9.0 database was produced and an all-versus-all BLASTP search was performed in parallel. The BLAST results for all isolates were combined and used as input into the pan-genome builder PanOCT[79] v.3.23 (-S Y -L 1 -M Y -H Y -V Y -N Y -F 1.33 -G y -c 0,25,50,75,100 -T). The resulting pan-genome was interrogated for the total number of protein-coding genes; a presence–absence matrix was constructed for each gene in all isolates. Correct construction of the pan-genome was assessed by mapping 55 previously identified *Af*293 essential genes[28] to the pan-genome using BLAST. Extraction of these gene sequences was performed using the R package biomaRt[80] v.2.44.4. Pan-genome clusters were then allocated a protein identifier by first mapping to the *Af*293 proteome with BLAST (-max_hsps 1 -perc_identity 70 -qcov_hsp_perc 50) then mapping to the RefSeq non-redundant database using DIAMOND[81] v.2.0.13 (-e 0.00001--query-cover 50--subject-cover 50--max-hsps 1--max-target-seqs 1--id 70); finally, conversion of protein identifier to gene accession was performed using Entrez Direct[82] v.1.6.2.

A custom script was used to extract pan-genome statistics. The R package micropan[83] v.2.1 was used to model the openness of the pan-genome using Heaps' law as described by Tettelin et al.[29], with the number of permutations set to 1,000. Data generated by this model were plotted with R v.4.0.2.

The metadata produced in this study were correlated with the gene variation matrix to identify genes associated with clade structure, itraconazole MIC values and *cyp51a* polymorphisms. Correlations were carried out using Scoary[84] v.1.6.16, supplied with the phylogenetic tree generated in this study. Associations were considered significant if the Bonferroni $P < 0.05$ and assigned to either binary

trait by odds ratio (OR). Subsequent Venn diagrams were produced using Eulerr[85] v.6.1.0. All parameters for the prior pipeline were default ones unless stated otherwise.

**Reporting Summary.** Further information on research design is available in the Nature Research Reporting Summary linked to this article.

## Data availability

All raw reads have been submitted to the European Nucleotide Archive under accession no. PRJEB27135. Six isolates (ARAF001-6) were sequenced as part of Abdolrasouli et al.[21], with reads deposited under accession no. PRJEB8623. Source data are provided with this paper.

## Code availability

Custom scripts for the pan-genome analysis can be found at https://github.com/harrychown/asp_pan.

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

## Acknowledgements

This study was partially supported by an unrestricted education grant from Gilead Sciences through their investigator sponsored research programme. J. Rhodes, T.S., A.P.B., P.S.D., D.A.J. and M.C.F. were supported by grants from the NERC (nos. NE/P001165/1 and NE/P000916/1), the UK Medical Research Council (MRC) (no. MR/R015600/1) and Wellcome Trust (no. 219551/Z/19/Z). D.A.J. is also funded by the Medical Research Council (grant no. MR/V037315/1) and Cystic Fibrosis Trust (grant no. SRC015). D.A.J. is funded by the Department of Health and Social Care (DHSC) Centre for Antimicrobial Optimisation (CAMO), Imperial College London. The views expressed in this publication are those of the authors and not necessarily those of the DHSC, National Health Service or National Institute for Health Research (NIHR). M.C.F. is a CIFAR Fellow in the Fungal Kingdom programme. K.D. was supported by a PhD studentship awarded by the School of Medicine, Trinity College Dublin. P.G.M. and J. Renwick (Dublin) received a project grant from the National Children's Hospital, Tallaght University Hospital, which in part supported this work. A.W. and E.B. are supported by the Wellcome Trust Strategic Award (grant no. 097377), MRC Centre for Medical Mycology (grant no. MR/N006364/2) at the University of Exeter and a Biotechnology and Biological Sciences Research Council EASTBIO grant (no. BB/M010996/1). The authors also acknowledge the Imperial College London Cystic Fibrosis Strategic Research Centre and NIHR CAMO.

## Author contributions

J. Rhodes and M.C.F. conceived and designed the study. The experimental work was carried out by A.A., T.S., Y.Z., K.D., A.T, A.P.B., R.B.P., A.F.T., A.S. and N.v.R. Isolates were supplied by R.C.B., A.W., E.B., P.L.W., T.R.R., N.G.M., P.G.M., S.H.C., E.M.J., P.D., S.S., D.A.J., J. Renwick and A.T.  J. Rhodes, H.C., M.B. and P.B. analysed the data and interpreted the results. J. Rhodes wrote the manuscript. All authors were involved in editing and reviewing the manuscript.

## Competing interests

The authors declare no competing interests.

## Additional information

**Extended data** is available for this paper at https://doi.org/10.1038/s41564-022-01091-2.

**Correspondence and requests for materials** should be addressed to Johanna Rhodes, Darius Armstrong-James or Matthew C. Fisher.

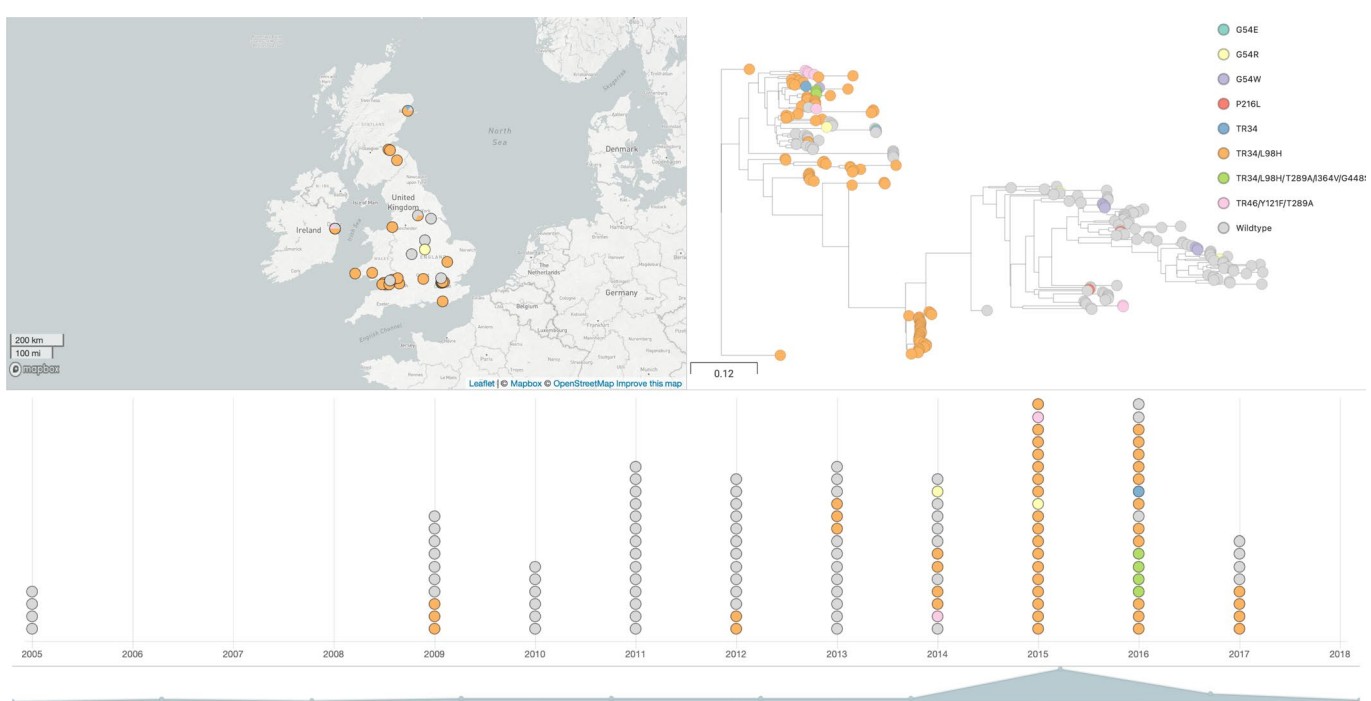

**Extended Data Fig. 1 |** Microreact project screenshot of the dataset https://microreact.org/project/viUDBzrCmTNKmY9Fu6Zhxi.

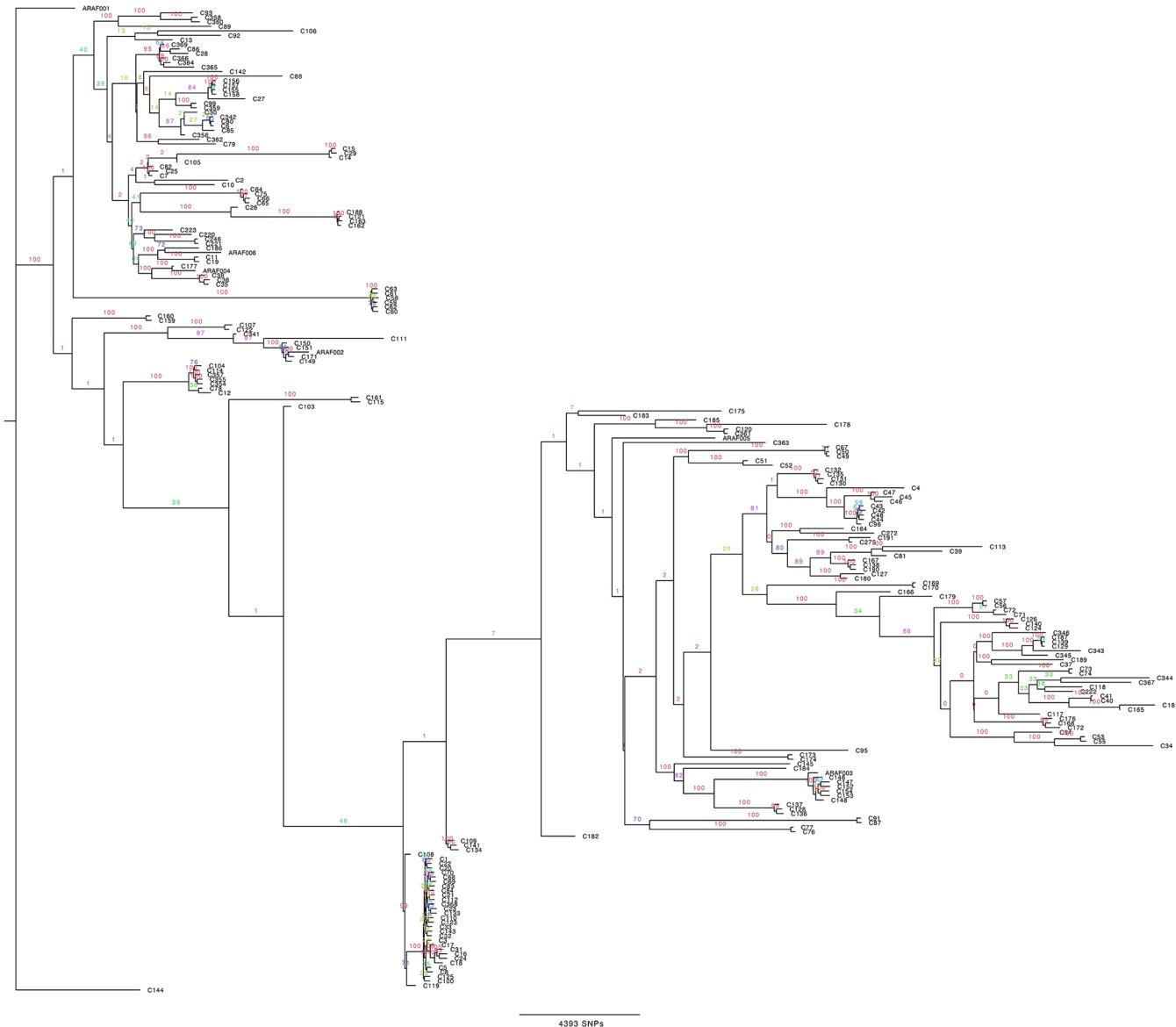

**Extended Data Fig. 2 |** Phylogenetic analysis of all 218 *A. fumigatus* isolates with bootstrap support over 1000 replicates performed on WGS SNP data to generate maximum-likelihood phylogeny. Branch lengths represent average number of SNPs.

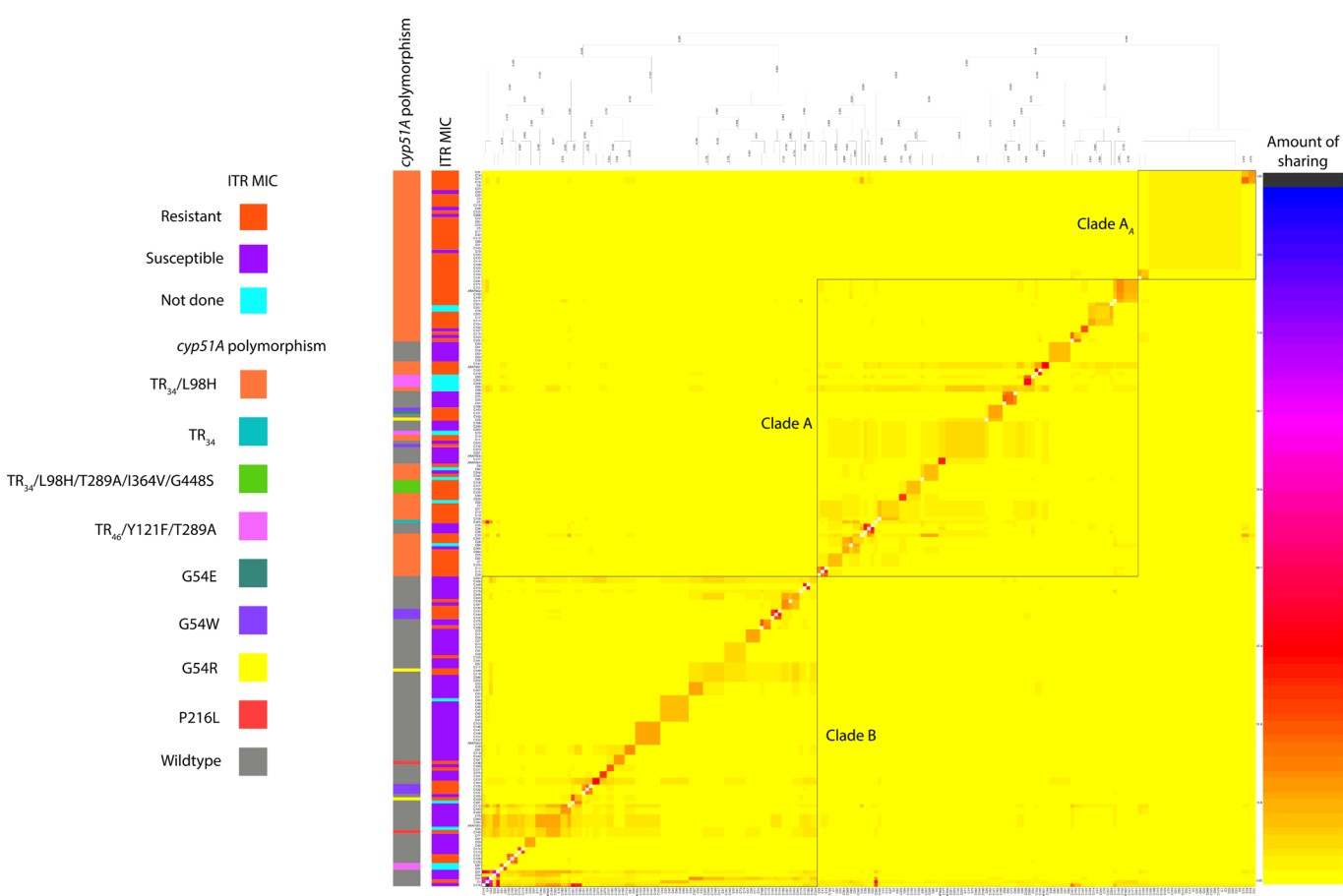

**Extended Data Fig. 3 | Genome sharing fineStructure analysis of *A. fumigatus* using genome-wide SNPs confirms the presence of three populations within the dataset.** Population-averaged coancestry matrix for the linked model dataset with associated *cyp51A* polymorphism and itraconazole MIC (defined as above or below 2 mg l⁻¹ for resistance or susceptibility, respectively, or not done). The right-hand scale bar represents the amount of genomic sharing, with blue/black representing the largest amount of sharing of genetic material and yellow representing the least amount of shared genetic material.

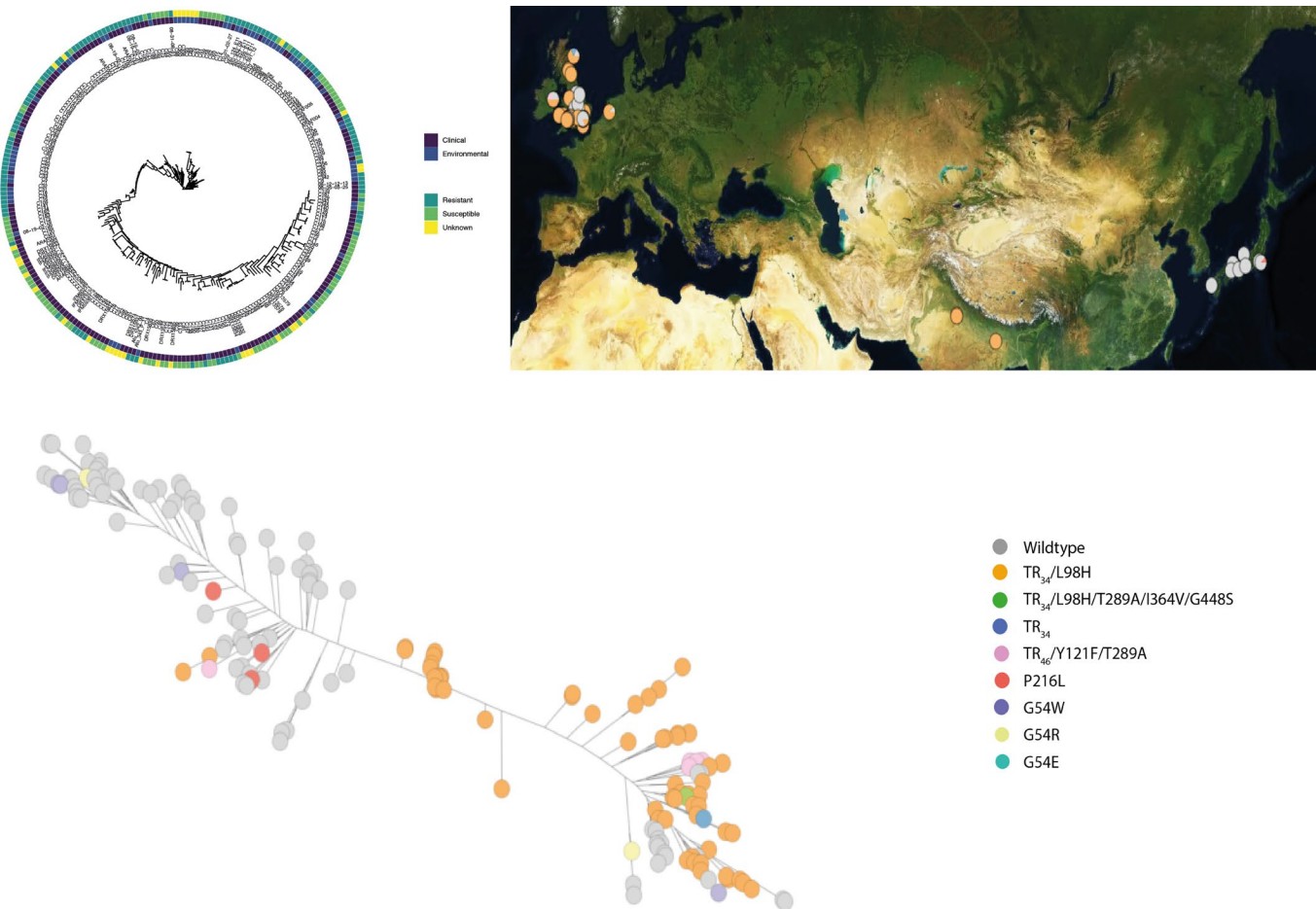

**Extended Data Fig. 4 |** Phylogenetic analysis of all 218 A. fumigatus isolates plus an additional 41 publicly available WGS of non-UK origin confirms the clade assignment into Clades A and B (Supplementary Fig. 6) was not an artifact of these data but also seen globally. These additional data comprised of 33 clinical and 8 environmental isolates. Maximum-likelihood phylogeny generated with bootstrap support over 1000 replicates on WGS SNP data.

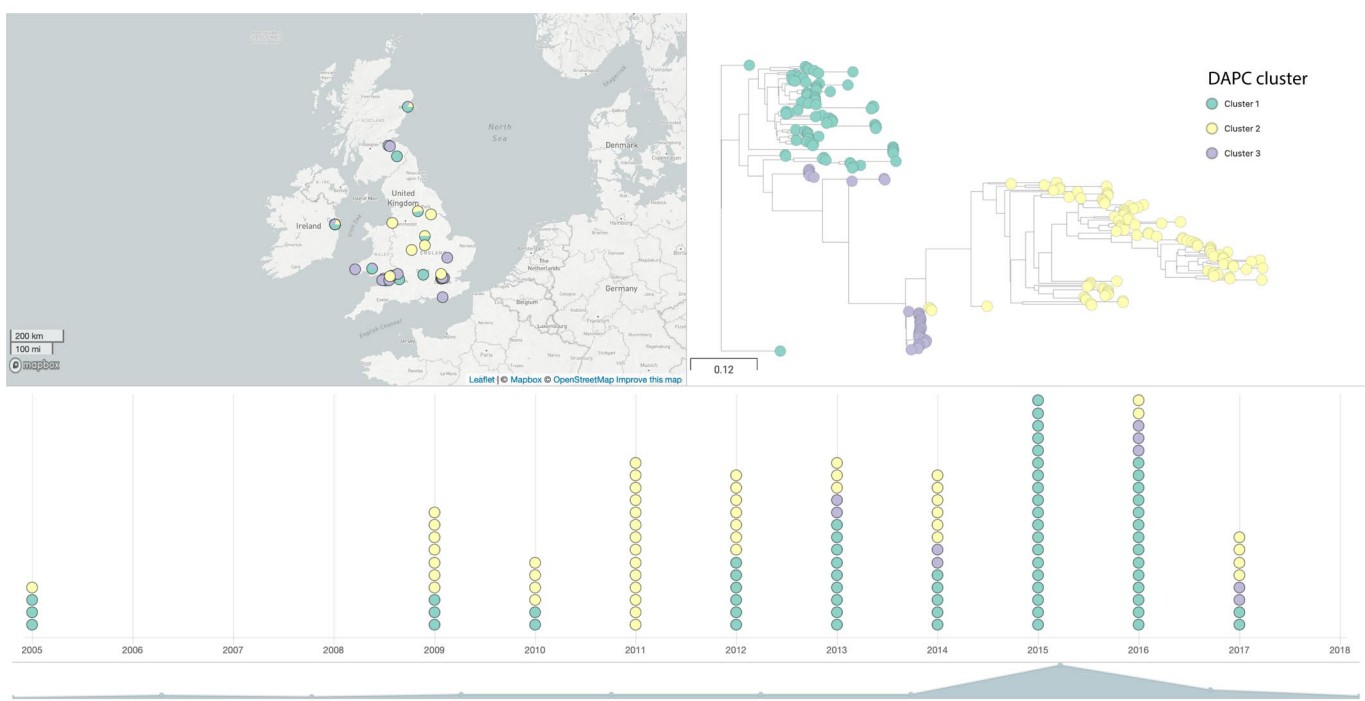

**Extended Data Fig. 5 |** Microreact project screenshot of the dataset with DAPC clusters showing lack of geographic and temporal clustering.

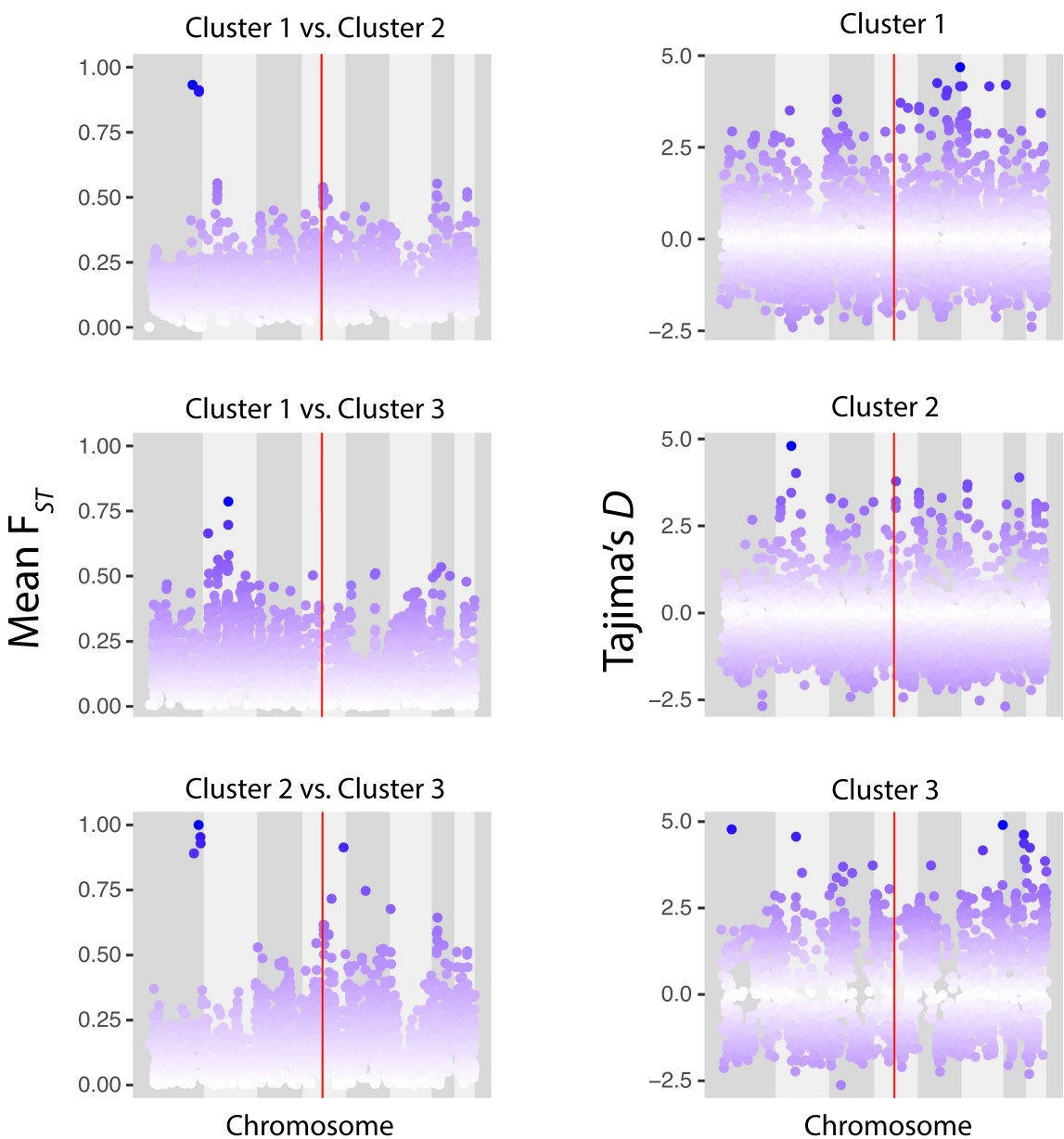

**Extended Data Fig. 6 |** Scatterplots of sliding 10-kb non-overlapping window estimates of $F_{ST}$ for each chromosome between isolates within Clusters 1 and 2, Clusters 1 and 3, and Clusters 2 and 3 (top to bottom, left panel). Scatterplots of Tajima's D estimates for each chromosome for all isolates within Clusters 1, 2 and 3 (right panel). The position of *cyp51A* is highlighted in red.

# Reporting Summary

Nature Research wishes to improve the reproducibility of the work that we publish. This form provides structure for consistency and transparency in reporting. For further information on Nature Research policies, see Authors & Referees and the Editorial Policy Checklist.

## Statistics

For all statistical analyses, confirm that the following items are present in the figure legend, table legend, main text, or Methods section.

| n/a | Confirmed | |
|---|---|---|
| ☐ | ☒ | The exact sample size (*n*) for each experimental group/condition, given as a discrete number and unit of measurement |
| ☐ | ☒ | A statement on whether measurements were taken from distinct samples or whether the same sample was measured repeatedly |
| ☐ | ☒ | The statistical test(s) used AND whether they are one- or two-sided *Only common tests should be described solely by name; describe more complex techniques in the Methods section.* |
| ☐ | ☒ | A description of all covariates tested |
| ☐ | ☒ | A description of any assumptions or corrections, such as tests of normality and adjustment for multiple comparisons |
| ☐ | ☒ | A full description of the statistical parameters including central tendency (e.g. means) or other basic estimates (e.g. regression coefficient) AND variation (e.g. standard deviation) or associated estimates of uncertainty (e.g. confidence intervals) |
| ☐ | ☒ | For null hypothesis testing, the test statistic (e.g. *F*, *t*, *r*) with confidence intervals, effect sizes, degrees of freedom and *P* value noted *Give P values as exact values whenever suitable.* |
| ☐ | ☒ | For Bayesian analysis, information on the choice of priors and Markov chain Monte Carlo settings |
| ☒ | ☐ | For hierarchical and complex designs, identification of the appropriate level for tests and full reporting of outcomes |
| ☒ | ☐ | Estimates of effect sizes (e.g. Cohen's *d*, Pearson's *r*), indicating how they were calculated |

*Our web collection on statistics for biologists contains articles on many of the points above.*

## Software and code

Policy information about availability of computer code

| Data collection | The whole genome sequence data were generated via the Illumina HiSeq sequencing platform |
|---|---|
| Data analysis | The WGS data were aligned to reference sequence using a custom pipeline as described in Rhodes et al 2015 (https://journals.asm.org/doi/abs/10.1128/mBio.00536-15) |

For manuscripts utilizing custom algorithms or software that are central to the research but not yet described in published literature, software must be made available to editors/reviewers. We strongly encourage code deposition in a community repository (e.g. GitHub). See the Nature Research guidelines for submitting code & software for further information.

## Data

Policy information about availability of data

All manuscripts must include a data availability statement. This statement should provide the following information, where applicable:
- Accession codes, unique identifiers, or web links for publicly available datasets
- A list of figures that have associated raw data
- A description of any restrictions on data availability

Source data are provided in the paper, its supplementary files, and a Source Data file for all Figures. The WGS data are deposited in the EBI-ENA under project accession PRJEB27135. Custom scripts for pangenome analysis can be found at github.com/harrychown/asp_pan

# Field-specific reporting

Please select the one below that is the best fit for your research. If you are not sure, read the appropriate sections before making your selection.

☒ Life sciences  ☐ Behavioural & social sciences  ☐ Ecological, evolutionary & environmental sciences

For a reference copy of the document with all sections, see nature.com/documents/nr-reporting-summary-flat.pdf

# Life sciences study design

All studies must disclose on these points even when the disclosure is negative.

| | |
|---|---|
| Sample size | Many isolates from both clinical and environmental sources were specifically selected for whole genome sequencing because they displayed phenotypic azole resistance (raised minimum inhibitory concentrations (MICs) to at least one triazole drug using EUCAST or CLSI) and do not constitute a randomised sample. |
| Data exclusions | Data from contaminated samples, e.g. contaminated seedlings and plates, were excluded. |
| Replication | Where multiple sequencing runs were used, a replicate of a single isolate was included to ensure no variation due to sequencing machine |
| Randomization | Samples were chosen randomly from each genotype per treatment per time point. |
| Blinding | Investigators were blinded with regards personal information and clinical data. |

# Reporting for specific materials, systems and methods

We require information from authors about some types of materials, experimental systems and methods used in many studies. Here, indicate whether each material, system or method listed is relevant to your study. If you are not sure if a list item applies to your research, read the appropriate section before selecting a response.

### Materials & experimental systems

| n/a | Involved in the study |
|---|---|
| ☒ | Antibodies |
| ☒ | Eukaryotic cell lines |
| ☒ | Palaeontology |
| ☒ | Animals and other organisms |
| ☒ | Human research participants |
| ☒ | Clinical data |

### Methods

| n/a | Involved in the study |
|---|---|
| ☒ | ChIP-seq |
| ☒ | Flow cytometry |
| ☒ | MRI-based neuroimaging |

## Antibodies

| | |
|---|---|
| Antibodies used | N/A |
| Validation | |

