## [Peer Review File · Nature Microbiology]

Peer Review Information

Journal: Nature Microbiology

Manuscript Title: Population genomics confirms acquisition of drug resistant *Aspergillus fumigatus* infection by humans from the environment

Corresponding author name(s): Dr Johanna Rhodes

Reviewer Comments & Decisions:

Decision Letter, initial version:

16th June 2021

Dear Jo,

Thanks for your initial responses to the reviewers requests. We ask that you address all reviewers concerns in full in revision. Thanks for confirming that you can carry out the ploidy analysis and the association between non-Af293 aspects of the genome and patient/environmental isolates. As for the queries about including genome from global datasets, what we believe would improve the generalisability of your findings would be an analysis of whether the two clades you report (... structured into two clades ('A' and 'B') with little interclade recombination and the majority of environmental azole resistance genetically clustered inside Clade A) is replicated in other independently generated datasets. This would enable others to relate the work you have done to the work being done around the world on establishing the spatial and molecular epidemiology and population genomics of *A. fumigatus* isolates. We also ask that you set the context by discussing what global surveys such as your previous work can show, and make it clear why detailed epidemiology studies with robustly phenotyped isolates are needed to being to unpick links between the environment and patients. It should be clear in the results section why you have chosen to phenotype under specific conditions and what limitations this imposes on understanding the findings.

Should further experimental data and text modifications allow you to address these criticisms, we would be happy to look at a revised manuscript.

2Please include a data availability statement as a separate section after Methods but before references, under the heading "Data Availability". This section should inform readers about the availability of the data used to support the conclusions of your study. This information includes accession codes to public repositories (data banks for protein, DNA or RNA sequences, microarray, proteomics data etc...), references to source data published alongside the paper, unique identifiers such as URLs to data repository entries, or data set DOIs, and any other statement about data availability. At a minimum, you should include the following statement: "The data that support the findings of this study are available from the corresponding author upon request", mentioning any restrictions on availability. If DOIs are provided, we also strongly encourage including these in the Reference list (authors, title, publisher (repository name), identifier, year). For more guidance on how to write this section please see:

<http://www.nature.com/authors/policies/data/data-availability-statements-data-citations.pdf>

- * If you have not done so already we suggest that you begin to revise your manuscript so that it conforms to our Article format instructions at <http://www.nature.com/nmicrobiol/info/final-submission>. Refer also to any guidelines provided in this letter.

When submitting the revised version of your manuscript, please pay close attention to our [href="https://www.nature.com/nature-research/editorial-policies/image-integrity">Digital Image Integrity Guidelines. and to the following points below:](https://www.nature.com/nature-research/editorial-policies/image-integrity)

- that unprocessed scans are clearly labelled and match the gels and western blots presented in figures.

- that control panels for gels and western blots are appropriately described as loading on sample processing controls

- all images in the paper are checked for duplication of panels and for splicing of gel lanes.

{Redacted}

Note: This url links to your confidential homepage and associated information about manuscripts you may have submitted or be reviewing for us. If you wish to forward this e-mail to co-authors, please delete this link to your homepage first.

Nature Microbiology is committed to improving transparency in authorship. As part of our efforts in this direction, we are now requesting that all authors identified as 'corresponding author' on published papers create and link their Open Researcher and Contributor Identifier (ORCID) with their account on the Manuscript Tracking System (MTS), prior to acceptance. This applies to primary research papers only. ORCID helps the scientific community achieve unambiguous attribution of all scholarly contributions. You can create and link your ORCID from the home page of the MTS by clicking on 'Modify my Springer Nature account'. For more information please visit www.springernature.com/orcid.

If you wish to submit a suitably revised manuscript we would hope to receive it within 6 months. If you cannot send it within this time, please let us know. We will be happy to consider your revision, even if a similar study has been accepted for publication at Nature Microbiology or published elsewhere (up to a maximum of 6 months).

Yours sincerely,

{Redacted}

Reviewer Expertise:

Referee #1: Fungal genomics, phylogenetics

Referee #2: Fungal pathogenesis, *A. fumigatus*, antifungal drug resistance

Referee #3: Antifungal drug resistance

Referee #4: Fungal pathogenesis, *A. fumigatus*

Reviewer Comments:

Reviewer #1 (Remarks to the Author):

In this study Rhodes et al investigate the population structure and evolutionary forces driving antifungal drug resistance in clinical and environmental strains of *Aspergillus fumigatus* from the UK and Ireland. *A. fumigatus* is a significant pathogen to humans, particularly the immunocompromised, those with underlying lung associated conditions and in particular a high level of incidence has been documented in cases associated with COVID-19. It's a very important pathogen deserving of serious attention and to my mind the study performed by Rhodes et al here fills in some of the information currently missing. Specifically they show that azole resistance can be linked to fungicidal use in

3agriculture. Furthermore non synonymous changes in genes associated with azole resistance are widely distributed and they have also found isolates with no known mechanism of resistance. Overall the manuscript is very well written and the methods and results are clear concise and sequential. Overall it is a great piece of work but I do have some comments I would like the authors to address.

1) I think it is true to say that this study is looking solely for SNPs associated with azole resistance. And it does locate these, some of them are novel and some of them are confirmatory. However when looking at all reads 93.4% of them map to the reference Af293 genome. It is known that individual *A. fumigatus* strains can have very different gene sets resulting in quite a dynamic pangenome. Previously published studies (and some currently under review) have shown that the accessory genome of *A. fumigatus* is ~15%. I wonder why the authors undertook a GWAS only approach when carrying out this analysis? Did they consider doing de novo assemblies and looking for novel genes (not found in Af293) that may be associated with azole resistance. I think it is something very worthwhile considering and will greatly strengthen the work undertaken here.

2) Following on from the point above, besides doing a GWAS study did the authors consider looking at levels of ploidy in their isolates. Is there evidence for chromosomal polymorphism?

Reviewer #2 (Remarks to the Author):

Rhodes et al. present a whole genome sequencing based analysis of ~200 *Aspergillus fumigatus* isolates collected across the United Kingdom and Republic of Ireland. The authors utilized CLSI and EUCAST methods to phenotype the majority of isolates for Triazole resistance and correlated genotype-phenotype associations with cutting edge bioinformatics analyses. Overall, the data generally support hypotheses already existing in the literature regarding the source of azole resistant *A. fumigatus* strains. Thus, the data overall are not surprising. The authors' data set does generate some new hypotheses to be tested regarding the genotype-phenotype association with triazole resistance and provide a robust data set for further investigations. My major critique of the study is the sole reliance on azole phenotyping. Ergosterol biosynthesis plays many important roles in fungal biology beyond responses to azoles. Additional phenotyping of the strains under conditions that perturb and/or rely on flux in ergosterol biosynthesis – may reveal alternative hypotheses that explain the emergence of the alleles discovered by the authors. In other words, the authors did not rigorously rule out other possible mechanisms for the emergence of these alleles – they go right to the “controversial” agriculture azole use hypothesis to explain their data, when in fact other environmental perturbations may indeed drive selection on the ergosterol biosynthesis pathway and associated genetic modifiers that indirectly leads to azole resistance in the clinic. I commend the author's on a very nice study – but the conclusions either need to be substantially modified to take into account alternative models – or the authors need to more rigorously phenotype the isolate collection to directly test alternative hypotheses. The emergence of drug resistance, in response to other environmental stresses, is well documented in bacteria and now also in fungi. This must be taken into consideration.

Reviewer #3 (Remarks to the Author):

4This paper describes a genome-based analysis of an *A. fumigatus* population in the UK and its relationship to the occurrence of azole resistance in both the environment and the clinic. The authors sampled over 218 isolates and among them identified azole resistance. The genome analysis revealed the presence of two major clades (Clades A and B). Interestingly clade A was containing most of the azole-resistant isolates, yet not only with different Cyp51A-dependent mechanisms but with still unknown mechanisms. In depth analysis of the Some clade A isolates also showed sub-clusters with little nucleotide divergence. The authors also observed by their analysis the close relationship between environmental and clinical isolates and thus confirmed the environmental route of azole resistance acquisition.

In addition, a genome-wide analysis ("GWAS") revealed the existence of genes associated with drug resistance, and among them Cyp51A as expected. Other population genetic conclusions were made. This paper nicely illustrates the power of systematic genome data analysis and brings also novel insights into the population structure of *A. fumigatus*. Some general comments/issues:

- 1) One of the highlights of the paper is the identification of 2 major clades, each of which differed by the occurrence of azole-resistant isolates. The analysis is based on UK isolates. Given that genome data are available from isolates with a non-UK origin, how is this clade assignment reflected in other available isolates? Such comparison data should be provided.
- 2) One particular finding was the identification of a TR34-only mutation in isolate C365. Strangely enough, the azole MICs of this isolate are higher than the associated classical TR34/L98H mutation (Table S2), which one would expect to be higher given the combination of 2 resistance markers. This is quite unexpected. How can this be explained?
- 3) L. 367-372: Here authors show that the 3 clusters "displayed no geographical or temporal clustering". Looking at the data available in Fig S3, it seems however that cluster 2 isolates were predominantly collected in 2011 and also mostly collected in central UK. Some clarifications are needed.
- 4) Fig 3 and lanes 403-404: "Individual isolates C4, C54 and C178 within Clade B displayed evidence for strong haplotype sharing". This type of data is hardly distinguishable in the current format. Some polishing is needed.
- 5) As a more general remark: even though the data are quite valuable, there are a lot of technical details in which the reading is difficult and thus dilute the main messages of the paper. This is OK for a specialist in population genetics but less accessible to the more general and major readership. There are several paragraphs which could be simplified or even removed:
 - a. Lanes 416-428: Too many details are given on the fixation indexes (FST). This will be valuable if only one knows what these index mean for the *A. fumigatus* population structure.
 - b. Lanes 430-432: What a Fst value of 0.5 implies for Chr 4 and 7 regions?
 - c. Lanes 445-458: The intra-clade detailed analysis needs to be simplified.
 - d. Lanes 539-543: the significance of IBD test could be removed.
- 6) L. 476: what is the function of Afu4g07010?
- 7) The 28 gene deletion set shows that only 2 genes impacted on the azole resistance phenotype (among them the known CYP51A locus). First, the result of *abcA* deletion should be commented, how can it explain the obtained phenotype? *abcA* deletion was already addressed in literature, how is this comparable to the current study? Second, the absence of phenotype for the remaining 26 loci rather tells a strong bottleneck between treewas associations and effective role of the loci in the azole resistance phenotype. This should be commented.

- 8) The discussion is too long and should again avoid too many details (as in lanes 651-674). It could be better stratified by including sub-headings, this would greatly help the capture of the interpretations. In addition, redundancies could be avoided (for example l. 731-745 in the close relationships between some clinical and environmental isolates).
- 9) Please check Fig numbering. For example Table S3 (l. 289) should be Table S2.

Reviewer #4 (Remarks to the Author):

Aspergillus fumigatus is a serious opportunistic pathogen that is difficult to treat for many reasons including an increasing number of antifungal resistant strains. There have been many studies addressing the occurrence of resistant strains at a population level, this study analyzes the occurrence of antifungal R in 218 isolates (153 being clinical isolates) obtained from the geographic location of England, Wales, Scotland and Republic of Ireland. I applaud the authors for rigor in analysis but they need to keep in mind that this is a small section of the world and it would be fruitful to compare this collection with those of other geographic areas. And temper conclusions accordingly.

The good news for the authors is that there are many sequences of other isolates from around the world. They need to bring in sequence data from other studies. Below are but a few. Some genomes overlap in manuscripts below which authors can figure this out. Will include clinical and environmental strains from different regions of the world and space station. Possibly some of these are in the study, I did not check thoroughly.

94 strains clinical and environmental = <https://www.ncbi.nlm.nih.gov/pmc/articles/PMC5082629/>

6 strains = <https://pubmed.ncbi.nlm.nih.gov/24486872/>

35 strains = <https://pubmed.ncbi.nlm.nih.gov/33705591/>

66 strains = <https://pubmed.ncbi.nlm.nih.gov/29149178/>

3 strains = <https://pubmed.ncbi.nlm.nih.gov/21876055/>

12 strains = <https://pubmed.ncbi.nlm.nih.gov/30714895/>

28 strains = <https://www.ncbi.nlm.nih.gov/pmc/articles/PMC6071029/>

8 Strains = <https://www.ncbi.nlm.nih.gov/pmc/articles/PMC4313286/>

1 strain Amazon = <https://www.frontiersin.org/articles/10.3389/fmicb.2018.01827/full>

Not sure one references this, it is BioRxiv (can one cite, I am not sure)

<https://www.biorxiv.org/content/10.1101/587402v2.full>

<https://www.biorxiv.org/content/10.1101/2021.04.19.440431v1>

Also, authors need to make sure they bring other relevant studies into their discussion such as:

<https://pubmed.ncbi.nlm.nih.gov/33234685/>

Author Rebuttal to Initial comments

6Reviewer Expertise:

Referee #1: Fungal genomics, phylogenetics

Referee #2: Fungal pathogenesis, *A. fumigatus*, antifungal drug resistance

Referee #3: Antifungal drug resistance

Referee #4: Fungal pathogenesis, *A. fumigatus*

Reviewer Comments:

Reviewer #1 (Remarks to the Author):

In this study Rhodes et al investigate the population structure and evolutionary forces driving antifungal drug resistance in clinical and environmental strains of *Aspergillus fumigatus* from the UK and Ireland. *A. fumigatus* is a significant pathogen to humans, particularly the immunocompromised, those with underlying lung associated conditions and in particular a high level of incidence has been documented in cases associated with COVID-19. It's a very important pathogen deserving of serious attention and to my mind the study performed by Rhodes et al here fills in some of the information currently missing. Specifically they show that azole resistance can be linked to fungicidal use in agriculture. Furthermore non synonymous changes in genes associated with azole resistance are widely distributed and they have also found isolates with no known mechanism of resistance. Overall the manuscript is very well written and the methods and results are clear concise and sequential. Overall it is a great piece of work but I do have some comments I would like the authors to address.

1) I think it is true to say that this study is looking solely for SNPs associated with azole resistance. And it does locate these, some of them are novel and some of them are confirmatory. However when looking at all reads 93.4% of them map to the reference Af293 genome. It is known that individual *A. fumigatus* strains can have very different gene sets resulting in quite a dynamic pangenome. Previously published studies (and some currently under review) have shown that the accessory genome of *A. fumigatus* is ~15%. I wonder why the authors undertook a GWAS only approach when carrying out this analysis? Did they consider doing de novo assemblies and looking for novel genes (not found in Af293) that may be associated with azole resistance. I think it is something very worthwhile considering and will greatly strengthen the work undertaken here.

We thank the reviewer for this suggestion; we have now included a pangenome analysis using all 218 isolates from this study, completed by Harry Chown under the supervision of Paul Bowyer (both have now been added as authors to the manuscript). New sections have been added (Methods lines 266-313, Results lines 624-658, Discussion lines 817-831) to cover this pangenome analysis, as well as Figure 5.

2) Following on from the point above, besides doing a GWAS study did the authors consider looking at levels of ploidy in their isolates. Is there evidence for chromosomal polymorphism?

Ploidy levels were checked in all isolates using bamCoverage in the deepTools 2.0 package, and only one isolate (ARAF005) was found to have a partial increase in ploidy on chromosome 1. This has been updated on lines 196-198 in the Methods, and lines 331-333 in Results.

Reviewer #2 (Remarks to the Author):

7Rhodes et al. present a whole genome sequencing based analysis of ~200 *Aspergillus fumigatus* isolates collected across the United Kingdom and Republic of Ireland. The authors' utilized CLSI and EUCAST methods to phenotype the majority of isolates for Triazole resistance and correlated genotype-phenotype associations with cutting edge bioinformatics analyses. Overall, the data generally support hypotheses already existing in the literature regarding the source of azole resistant *A. fumigatus* strains. Thus, the data overall are not surprising. The authors' data set does generate some new hypotheses to be tested regarding the genotype-phenotype association with triazole resistance and provide a robust data set for further investigations. My major critique of the study is the sole reliance on azole phenotyping. Ergosterol biosynthesis plays many important roles in fungal biology beyond responses to azoles. Additional phenotyping of the strains under conditions that perturb and/or rely on flux in ergosterol biosynthesis – may reveal alternative hypotheses that explain the emergence of the alleles discovered by the authors. In other words, the authors did not rigorously rule out other possible mechanisms for the emergence of these alleles – they go right to the “controversial” agriculture azole use hypothesis to explain their data, when in fact other environmental perturbations may indeed drive selection on the ergosterol biosynthesis pathway and associated genetic modifiers that indirectly leads to azole resistance in the clinic. I commend the author's on a very nice study – but the conclusions either need to be substantially modified to take into account alternative models – or the authors need to more rigorously phenotype the isolate collection to directly test alternative hypotheses. The emergence of drug resistance, in response to other environmental stresses, is well documented in bacteria and now also in fungi. This must be taken into consideration.

We have added to the discussion the model that unstable, epigenetic-driven resistance could be acquired, rather than via the agricultural application of azole fungicides in lines 831-834. However, given there are studies (e.g. Barber et al. 2020 mBio) that show resistance in horticulture (flower cultivation etc), it is still possible that this hypothesis is not 'controversial'. Indeed, given that we show indistinguishable environmental and clinical genotypes, with the environmental source being a soil sample, we cannot rule out the agriculture azole use driving selection.

Reviewer #3 (Remarks to the Author):

This paper describes a genome-based analysis of an *A. fumigatus* population in the UK and its relationship to the occurrence of azole resistance in both the environment and the clinic. The authors sampled over 218 isolates and among them identified azole resistance. The genome analysis revealed the presence of two major clades (Clades A and B). Interestingly clade A was containing most of the azole-resistant isolates, yet not only with different Cyp51A-dependent mechanisms but with still unknown mechanisms. In depth analysis of the Some clade A isolates also showed sub-clusters with little nucleotide divergence. The authors also observed by their analysis the close relationship between environmental and clinical isolates and thus confirmed the environmental route of azole resistance acquisition.

In addition, a genome-wide analysis (“GWAS”) revealed the existence of genes associated with drug resistance, and among them Cyp51A as expected. Other population genetic conclusions were made.

This paper nicely illustrates the power of systematic genome data analysis and brings also novel insights into the population structure of *A. fumigatus*. Some general comments/issues:

1) One of the highlights of the paper is the identification of 2 major clades, each of which differed by the occurrence of azole-resistant isolates. The analysis is based on UK isolates. Given that genome data are available from isolates with a non-UK origin, how is this clade assignment reflected in other available isolates? Such comparison data should be provided.

8An additional 41 isolates from publicly available WGS data of non-UK origin were added to the phylogenetic analyses to confirm the clade assignment (Figure S6, lines 409-412). Therefore, the clade assignment is seen globally and not an artefact of these data. In addition, there are now multiple studies which have used the Clade A and B nomenclature to describe the difference in azole-resistant isolates from across the globe, showing this is not a UK-only phenomena, and this is also reflected in the discussion. This is also discussed on lines 703-706.

2) One particular finding was the identification of a TR34-only mutation in isolate C365. Strangely enough, the azole MICs of this isolate are higher than the associated classical TR34/L98H mutation (Table S2), which one would expect to be higher given the combination of 2 resistance markers. This is quite unexpected. How can this be explained?

The MIC of the TR34-only isolate (C365) was 16 for itraconazole and 2 for voriconazole. We do not see this an anomalous, as there are a number of isolates containing TR34/L98H that had MICs >16 for itraconazole and MIC of 2 or higher for voriconazole (e.g. C159, C160, C143, C144, C133, C134, C125, C122, C123, C100, C83, C28, C29, C30, C31, C20-C25, C1-C18).

3) L. 367-372: Here authors show that the 3 clusters “displayed no geographical or temporal clustering”. Looking at the data available in Fig S3, it seems however that cluster 2 isolates were predominantly collected in 2011 and also mostly collected in central UK. Some clarifications are needed.

There is 1 isolate in cluster 2 collected in 2005, 7 in 2009, 4 in 2010, 14 isolates collected in 2011, 7 in 2012, 3 in 2013, 6 in 2014, 41 in 2015, 9 in 2016, 4 in 2017. All isolates within cluster 2 are representative of the geographical sampling presented within these data – however, the majority were collected within the Greater London area. These London isolates were collected between 2015 and 2017, so we still believe there is no geographical or temporal clustering within the DAPC clusters presented.

4) Fig 3 and lanes 403-404: “Individual isolates C4, C54 and C178 within Clade B displayed evidence for strong haplotype sharing”. This type of data is hardly distinguishable in the current format. Some polishing is needed.

Zooming in on Figure 3 provides the relevant information

5) As a more general remark: even though the data are quite valuable, there are a lot of technical details in which the reading is difficult and thus dilute the main messages of the paper. This is OK for a specialist in population genetics but less accessible to the more general and major readership. There are several paragraphs which could be simplified or even removed:

a. Lanes 416-428: Too many details are given on the fixation indexes (FST). This will be valuable if only one knows what these index mean for the *A. fumigatus* population structure.

Fixation index is a general population genomics term, not specific for *Aspergillus fumigatus*. However, clarification on F_{ST} has been added on lines 474-477.

b. Lanes 430-432: What a F_{ST} value of 0.5 implies for Chr 4 and 7 regions?

Clarification added line 490.

c. Lanes 445-458: The intra-clade detailed analysis needs to be simplified.

The authors feel this is well explained already.

d. Lanes 539-543: the significance of IBD test could be removed.

No – this result, whilst not significant, shows that the Clade A_A clone is likely to be found outside of the British Isles and Republic of Ireland, which is an important finding.

6) L. 476: what is the function of Afu4g07010?

It has no known function

7) The 28 gene deletion set shows that only 2 genes impacted on the azole resistance phenotype (among them the known CYP51A locus). First, the result of *abcA* deletion should be commented, how can it explain the obtained phenotype? *abcA* deletion was already addressed in literature, how is this comparable to the current study? Second, the absence of phenotype for the remaining 26 loci rather tells a strong bottleneck between tree was associations and effective role of the loci in the azole resistance phenotype. This should be commented.

Comment on *abcA* added lines 810-811. However, the following sentence does state that this is purely a confirmatory approach to show that GWAS is able to detect loci associated with resistance – and it picked up *cyp51a* and *abcA*, which are known (and therefore should be picked up). We highlight that this sort of reverse genomics approach is crucial to understand drug resistance in *A. fumigatus* further; it's not an attempt to compare with other studies.

8) The discussion is too long and should again avoid too many details (as in lanes 651-674). It could be better stratified by including sub-headings, this would greatly help the capture of the interpretations. In addition, redundancies could be avoided (for example I. 731-745 in the close relationships between some clinical and environmental isolates).

Nature Microbiology articles do not tend to have subheadings in the discussion. The authors feel that trimming the discussion down would lose a lot of the detail that this reviewer and other reviewers have asked for. However, we have tried to make it more concise, where appropriate.

9) Please check Fig numbering. For example Table S3 (l. 289) should be Table S2.

Updated

Reviewer #4 (Remarks to the Author):

Aspergillus fumigatus is a serious opportunistic pathogen that is difficult to treat for many reasons including an increasing number of antifungal resistant strains. There have been many studies addressing the occurrence of resistant strains at a population level, this study analyzes the occurrence of antifungal R in 218 isolates (153 being clinical isolates) obtained from the geographic location of England, Wales, Scotland and Republic of Ireland. I applaud the authors for rigor in analysis but they need to keep in mind that this is a small section of the world and it would be fruitful to compare this collection with those of other geographic areas. And temper conclusions accordingly.

The good news for the authors is that there are many sequences of other isolates from around the world. They need to bring in sequence data from other studies. Below are but a few. Some genomes overlap in manuscripts below which authors can figure this out. Will include clinical and environmental strains from different regions of the world and space station. Possibly some of these are in the study, I did not check thoroughly.

94 strains clinical and environmental = <https://www.ncbi.nlm.nih.gov/pmc/articles/PMC5082629/>

6 strains = <https://pubmed.ncbi.nlm.nih.gov/24486872/>

35 strains = <https://pubmed.ncbi.nlm.nih.gov/33705591/>

66 strains = <https://pubmed.ncbi.nlm.nih.gov/29149178/>

3 strains = <https://pubmed.ncbi.nlm.nih.gov/21876055/>

12 strains = <https://pubmed.ncbi.nlm.nih.gov/30714895/>

28 strains = <https://www.ncbi.nlm.nih.gov/pmc/articles/PMC6071029/>

8 Strains = <https://www.ncbi.nlm.nih.gov/pmc/articles/PMC4313286/>

1 strain Amazon = <https://www.frontiersin.org/articles/10.3389/fmicb.2018.01827/full>

Not sure one references this, it is BioRxiv (can one cite, I am not sure)

<https://www.biorxiv.org/content/10.1101/587402v2.full>

<https://www.biorxiv.org/content/10.1101/2021.04.19.440431v1>

As mentioned in response to Reviewer #3, 41 additional isolates that are publicly available have been added to the phylogenetic analysis to confirm the clade structure

Also, authors need to make sure they bring other relevant studies into their discussion such as:

<https://pubmed.ncbi.nlm.nih.gov/33234685/>

11This study has been added (line 834)

Decision Letter, first revision:

Our ref: NMICROBIOL-21041097A

18th January 2022

Dear Dr. Rhodes,

Thank you for submitting your revised manuscript "Tracing patterns of evolution and acquisition of drug resistant *Aspergillus fumigatus* infection from the environment using population genomics" (NMICROBIOL-21041097A). It has now been seen by two of the original referees and their comments are below. The reviewers find that the paper has improved in revision, and therefore we'll be happy in principle to publish it in Nature Microbiology, pending minor revisions to comply with our editorial and formatting guidelines.

Thank you again for your interest in Nature Microbiology Please do not hesitate to contact me if you have any questions.

Sincerely,
{Redacted}

Reviewer #3 (Remarks to the Author):

The authors addressed throughout all issues with corresponding answers and manuscript modifications.

Reviewer #4 (Remarks to the Author):

12I am satisfied

Decision Letter Final Checks

Our ref: NMICROBIOL-21041097A

1st February 2022

Dear Jo,

Thank you for your patience as we've prepared the guidelines for final submission of your Nature Microbiology manuscript, "Tracing patterns of evolution and acquisition of drug resistant *Aspergillus fumigatus* infection from the environment using population genomics" (NMICROBIOL-21041097A). Please carefully follow the step-by-step instructions provided in the attached file, and add a response in each row of the table to indicate the changes that you have made. Please also check and comment on any additional marked-up edits we have proposed within the text. Ensuring that each point is addressed will help to ensure that your revised manuscript can be swiftly handed over to our production team.

In recognition of the time and expertise our reviewers provide to Nature Microbiology's editorial process, we would like to formally acknowledge their contribution to the external peer review of your manuscript entitled "Tracing patterns of evolution and acquisition of drug resistant *Aspergillus fumigatus* infection from the environment using population genomics". For those reviewers who give their assent, we will be publishing their names alongside the published article.

13Nature Microbiology offers a Transparent Peer Review option for new original research manuscripts submitted after December 1st, 2019. As part of this initiative, we encourage our authors to support increased transparency into the peer review process by agreeing to have the reviewer comments, author rebuttal letters, and editorial decision letters published as a Supplementary item. When you submit your final files please clearly state in your cover letter whether or not you would like to participate in this initiative. Please note that failure to state your preference will result in delays in accepting your manuscript for publication.

Cover suggestions

As you prepare your final files we encourage you to consider whether you have any images or illustrations that may be appropriate for use on the cover of Nature Microbiology.

Nature Microbiology has now transitioned to a unified Rights Collection system which will allow our Author Services team to quickly and easily collect the rights and permissions required to publish your work. Approximately 10 days after your paper is formally accepted, you will receive an email in providing you with a link to complete the grant of rights. If your paper is eligible for Open Access, our Author Services team will also be in touch regarding any additional information that may be required to arrange payment for your article.

Please note that *Nature Microbiology* is a Transformative Journal (TJ). Authors may publish their research with us through the traditional subscription access route or make their paper immediately open access through payment of an article-processing charge (APC). Authors will not be required to make a final decision about access to their article until it has been accepted. [Find out more about Transformative Journals](https://www.springernature.com/gp/open-research/transformative-journals)

Authors may need to take specific actions to achieve compliance with funder and institutional open access mandates. For submissions from January 2021, if your research is supported by a funder that requires immediate open access (e.g. according to [Plan S principles](https://www.springernature.com/gp/open-research/plan-s-compliance)) then you should select the gold OA route, and we will direct you to the compliant route where possible. For authors selecting the subscription publication route our standard licensing terms will need to be accepted, including our [self-archiving policies](https://www.springernature.com/gp/open-research/policies/journal-policies). Those standard licensing terms will supersede any other terms that the author or any third party may assert apply to any version of the manuscript.

Please use the following link for uploading these materials:
{Redacted}

Best regards,
{Redacted}

Reviewer #3:

Remarks to the Author:

The authors addressed throughout all issues with corresponding answers and manuscript modifications.

Reviewer #4:

Remarks to the Author:

I am satisfied

Final Decision Letter:

Dear Dr Rhodes,

I am pleased to accept your Article "Population genomics confirms acquisition of drug resistant *Aspergillus fumigatus* infection by humans from the environment" for publication in Nature Microbiology. Thank you for having chosen to submit your work to us and many congratulations.

Acceptance of your manuscript is conditional on all authors' agreement with our publication policies (see <https://www.nature.com/nmicrobiol/editorial-policies>). In particular your manuscript must not be published elsewhere and there must be no announcement of the work to any media outlet until the publication date (the day on which it is uploaded onto our website).

16Please note that *Nature Microbiology* is a Transformative Journal (TJ). Authors may publish their research with us through the traditional subscription access route or make their paper immediately open access through payment of an article-processing charge (APC). Authors will not be required to make a final decision about access to their article until it has been accepted. [Find out more about Transformative Journals](https://www.springernature.com/gp/open-research/transformative-journals)

Authors may need to take specific actions to achieve compliance with funder and institutional open access mandates. For submissions from January 2021, if your research is supported by a funder that requires immediate open access (e.g. according to [Plan S principles](https://www.springernature.com/gp/open-research/plan-s-compliance)) then you should select the gold OA route, and we will direct you to the compliant route where possible. For authors selecting the subscription publication route our standard licensing terms will need to be accepted, including our [self-archiving policies](https://www.springernature.com/gp/open-research/policies/journal-policies). Those standard licensing terms will supersede any other terms that the author or any third party may assert apply to any version of the manuscript.

To assist our authors in disseminating their research to the broader community, our SharedIt initiative provides you with a unique shareable link that will allow anyone (with or without a subscription) to read the published article. Recipients of the link with a subscription will also be able to download and

17print the PDF.
